# Development and Applications of Somatic Embryogenesis in Grapevine (*Vitis* spp.)

**DOI:** 10.3390/plants13223131

**Published:** 2024-11-07

**Authors:** Angela Carra, Akila Wijerathna-Yapa, Ranjith Pathirana, Francesco Carimi

**Affiliations:** 1Istituto di Bioscienze e BioRisorse (IBBR), Consiglio Nazionale delle Recerche, Via Ugo La Malfa 153, 90146 Palermo, Italy; angela.carra@ibbr.cnr.it (A.C.); francesco.carimi@ibbr.cnr.it (F.C.); 2School of Biological Sciences, The University of Queensland, St Lucia, QLD 4072, Australia; akila.ymw@gmail.com; 3School of Agriculture, Food and Wine, University of Adelaide, Waite Campus Research Precinct—S120, Main Waite Bldg., Waite Rd, Urrbrae, SA 5064, Australia

**Keywords:** germplasm, mutagenesis, in vitro culture, tissue culture, propagation, conservation, somaclonal variation, transformation, chimera

## Abstract

Somatic embryogenesis (SE) provides alternative methodologies for the propagation of grapevine (*Vitis* spp.) cultivars, conservation of their germplasm resources, and crop improvement. In this review, the current state of knowledge regarding grapevine SE as applied to these technologies is presented, with a focus on the benefits, challenges, and limitations of this method. The paper provides a comprehensive overview of the different steps involved in the grapevine SE process, including callus induction, maintenance of embryogenic cultures, and the production of plantlets. Additionally, the review explores the development of high-health plant material through SE; the molecular and biochemical mechanisms underlying SE, including the regulation of gene expression, hormone signaling pathways, and metabolic pathways; as well as its use in crop improvement programs. The review concludes by highlighting the future directions for grapevine SE research, including the development of new and improved protocols, the integration of SE with other plant tissue culture techniques, and the application of SE for the production of elite grapevine cultivars, for the conservation of endangered grapevine species as well as for cultivars with unique traits that are valuable for breeding programs.

## 1. Introduction

The lack of motility and the resulting inability of plants to escape from predators, parasites, and changes in the surrounding environment has led to the development of very efficient defense strategies. Plants are generally highly plastic organisms compared with animals, being able to modulate their development depending on endogenous and environmental signals, even reprogramming the fate of somatic cells. At the basis of this ability is the mechanism of totipotency that is observed in plant cells [1,2]. Cell fate reprogramming is complex and frequently associated with significant changes in chromatin status. Chromatin change is characterized by DNA methylation and histone chemical modifications, mainly methylation or acetylation [3,4]. The early observations of the capacity that plants have to react to tissue injury by leading partially differentiated somatic cells to change their fate, thus favoring the formation of an unorganized cell mass, called a callus, which plays a prominent role in damaged tissue but is also capable of regenerating new organs, led to the pioneering studies of plant tissue and cell culture in vitro. The first theoretical basis for plant tissue culture dates back to the early 1900s when Gottlieb Haberlandt [5] observed that the cells of plant tissues cultured in vitro survived and increased in volume. However, due to a lack of adequate culture medium containing plant growth regulators (PGRs) such as 3-indole-acetic acid (IAA—isolated in 1885 by the chemist Salkowski), he failed to observe cell division under the experimental conditions used in his study [6]. Haberlandt hypothesized that a single cell is a living unit, an individual in itself that is to some extent independent of the whole organism: ‘*Als Elementar Organismus …… ist die Zelle eine Lebenseinheit, ein Individuum fiir sich, das ein vom Gesamtorganismus his zu einem gewissen Grade unabhangiges Eigenleben fuhrt*’ [7]. This led to the idea that a single cell could be capable of giving rise to a complete and functional plant [8]. Direct evidence supporting the hypothesis that it was possible to regenerate plant organs in vitro was lacking until the end of the 1950s. Three independent groups, between 1957 and 1958, discovered the regeneration process in *Oenanthe aquatica* [9] and *Daucus carota* [10,11]. It was Reinert [10] who used the term ‘adventive embryos’ for the first time in 1959. Finally, Haberlandt’s hypothesis of producing a whole plant from a single cell was demonstrated in the mid-1960s in two publications as part of Vilma Vasil’s doctoral thesis under the direction of Hildebrandt at the University of Wisconsin, thus demonstrating the totipotency of plant cells [12,13].

The dedifferentiation of plant cells has long attracted interest as a key process for understanding the plasticity of plant development. These studies led to the hypothesis that many mature plant cells retain totipotency and related dedifferentiation to the initial step of the expression of totipotency. Even though all the diploid cells in an individual have the same genomic DNA, different cell types have distinct cell characteristics, and only some cells are totipotent to become an embryo. This discrepancy suggests that differences in the ability to generate somatic embryos are not driven simply by DNA sequences but probably by different epigenetic changes at different loci of totipotent cells. In fact, totipotent cells differ from their surrounding somatic cells mainly in five respects: a large nucleus, a large nucleolus, fragmented vacuoles, symplasmic isolation, and low levels of heterochromatin [2,14,15]. Recent studies have shown that epigenetic aspects related to chromatin remodeling play a key role in SE and can serve as good markers [3,16,17].

Considerable progress was achieved after the discovery of the hormonal control of cell proliferation and organogenesis in vitro in the 1950s [18]. These studies allowed the identification of efficient in vitro regeneration protocols based on organogenesis and SE. If somatic cells are stimulated to generate cells with embryogenic potential, the new cells can give rise to structures capable of regenerating a complete plant [17,19].

SE is a process by which plants can produce bipolar structures from a single somatic cell without meiosis and fertilization; therefore, the new plant derived from a somatic embryo is genetically identical to the mother plant. This complex process can follow two paths, called direct (from a single somatic cell) and indirect (from undifferentiated cells) embryogenesis. However, it is difficult to distinguish between the two routes, which can sometimes occur simultaneously from the same explant. The most common route is indirect SE and begins through the typical formation of a callus, an apparently disorganized mass of cells showing varying degrees of compactness, with many examples in grapevine [20,21,22,23,24,25,26,27]. During this process of dedifferentiation and differentiation of plant cells, the explant responds not only to endogenous but also to exogenous stimuli (including different types of stress), which modify the endogenous hormonal balance. The evidence supports the notion of a major role for auxins in the establishment of polarity and embryo initiation and development [28,29,30,31,32]. For the understanding of this important plant regeneration model, the interactions between the different PGRs, mainly auxins, cytokinins (CKs), ethylene, and abscisic acid (ABA), during the induction of SE are of fundamental importance [33]. In particular, it has been observed that the induction of the auxin biosynthesis genes *TAA1*/*TAR2*, an increase in cellular auxin concentration, and its polar transport are required for cell reprogramming and embryo regeneration [29,30].

SE has allowed the development of an increasing number of practical and scientific applications. For example, it has the potential for the genetic and sanitary improvement of genotypes of commercial importance, as well as providing insights into the underlying mechanisms of biological processes [34,35,36,37,38,39]. Furthermore, the application of the most modern Clustered Regularly Interspaced Short Palindromic Repeats (CRISPR)-derived biotechnologies that have revolutionized the genetic engineering field is limited in many crops by the lack of efficient in vitro plant regeneration protocols [40], and so far in grapevine this has been achieved using SE [41,42,43,44,45,46]. Finally, the preservation of germplasm through SE is an efficient method of conservation at reduced cost for those species which cannot be propagated through seeds [47]. Our purpose is to provide a comprehensive and updated overview of the application of SE in grapevine as well as critically discuss and highlight the future perspectives and challenges.

## 2. Explant Type as a Major Factor Determining the Success of Somatic Embryogenesis in Grapevine

One of the main constraints influencing the different applications of in vitro SE as a tool for plant regeneration is the low embryogenic potential of many crops and genotypes. A plant species/genotype, a tissue, or a developmental phase of a plant is termed recalcitrant if commonly used protocols fail to regenerate somatic embryos in vitro [48]. This recalcitrance affects not only embryo differentiation but also the subsequent steps in the regeneration process from embryo germination to plantlet acclimatization in vivo. Although SE can be induced from a range of tissues (Table 1), the correct choice of the type of explant is of fundamental importance [49]. In fact, it is important to determine first which part of the plant contains the most responsive tissues and at what stage of development and time of the year they must be collected. The age of the cells is also important in different species; usually, younger cells are reported as those in the most responsive state to induce embryogenic cultures [50,51,52].

In grapevine, the best results are usually obtained with explants of floral origin, such as whole flowers, anthers, filaments, stigmas/styles, ovaries, and pistils (Figure 1A,B; Table 1). The presence of multiple pathways of auxin biosynthesis within the inflorescence makes this organ an elegant model for studying SE. High-precision dynamic spatiotemporal auxin gradients within the inflorescence meristem are coordinated with growth [53,54]. This ensures that cells are exposed to a high level of auxin over time to activate organogenesis. As floral meristems are initiated in the axils, the timing and duration of exposure of cells to high auxin concentrations is governed temporally within the tissue [54]. This makes the floral organs very sensitive to the presence of exogenous PGRs. The other key factor is epigenetic modifications. During the transition from the vegetative to the reproductive phase of development, the patterns of DNA and histone methylation change significantly [55]. Thus, not only the type and concentration of exogenous PGR, but also the timing of harvest of explants to initiate cultures is critical for the success of grapevine SE.

**Table 1 plants-13-03131-t001:** Successful somatic embryogenesis protocols in *Vitis* spp.

Applications	PGRs	Explant Types	Reference
Confirmation of chimerism	2,4-D 4.5 μM+ BA 8.9 μM	Ovary, anthers/filaments	[56]
Biomass production	NAA 0.5 μM + BA 2.2 μM	Tender stems	[57]
Chemical mutagenesis	2,4-D 4.5 µM + BA8.9 µM; NOA 5.0 µM + BA 9.0 µM	Flower buds	[58]
Cryopreservation	NOA 5 μM+ BA 2 μM	Anthers	[59]
Cryopreservation	NOA 5 μM + BA 1 μM	Anthers	[60]
Establishment of protocol	2,4-D 6.5 µM + BA 7 μM	Immature anthers	[61]
Establishment of protocol	NOA 5 µM + BA 0.9–4.5 µM	Leaves, anthers	[62]
Establishment of protocol	NOA 5 µM + BA 9 µM	Styles/stigmas	[63]
Establishment of protocol	NOA 9.9 µM + BA 4.5 µM; BA 9 µM	Styles/stigmas	[51]
Establishment of protocol	NAA 0.4 µM + BA 10 µM + GA3 2.8 µM	Tendrils	[64]
Establishment of protocol	2,4-D 4.5 µM +BA 9 µM	Whole flowers, anthers, ovary	[65]
Establishment of protocol	2,4-D 4.5 µM + BA 9 µM	Whole flowers, anthers, ovary	[66]
Establishment of protocol	2,4-D 9 µM + TDZ 11.35 µM	Anthers	[67]
Factor interaction to overcome dormancy	2,4-D 4.4 μM + BA 4.4 μM	Anthers	[68]
Factor interaction, assessment of genetic stability of SE-derived plants	2,4-D 5 μM + CPPU 5 μM; NOA 20 μM + TDZ 4 μM; NOA5 μM + BA4.4 μM; NOA 10 μM+ BA 4.4 μM	Ovary, anther/filament, stigmas/styles	[47]
Factor interaction	2,4-D 9.0 μM+ BA 4.4 μM	Unopened and fully opened leaves, petioles	[69]
Genetic transformation for disease resistance	NOA 5 μM + BA 4.44 μM + phenylalanine 5.0 mM	In vitro leaves	[70]
Genetic transformation	2,4-D 4.5 µM + BA 9 µM	Anthers	[71]
Genetic transformation	2,4-D + BA + picloram several concentrations	Anthers, ovaries, flower buds	[72]
Genetic transformation	2,4-D 4.52 µM + BA 4.4 µM: 2,4-D 4.52 µM + NOA 2.5 µM + 4-CPPU 5 µM	Floral explants	[73]
Genetic transformation	BA 8.9 μM + 2,4-D 4.5 μM BA 4.5 μM + 2,4-D 5 μM	Whole flowers, anther/filament, pistils	[74]
Improvement of plant conversion efficiency	NOA 2.5 μM + 2,4-D 2.3 μM + 4-CPPU 4 μM	Whole flower bud	[75]
Improvement of protocols for recalcitrant genotypes	2,4-D 4.5 µM + BA 8.9 µM	Immature inflorescences	[37]
Improvement of regeneration protocol	BA 8.9 μM + 2,4-D 4.5 μM BA 4.5 μM + 2,4-D 5 μM	Whole flowers, anther/filament, pistils	[76]
Long-term SE	2,4-D 5 µM + BA 1 µM	Anthers, ovary	[77]
Long-term SE	2,4-D 2.5 µM + NOA 2.5 µM + 4-CPPU 5 µM	Anthers, ovary	[78]
Long-term SE	2,4-D 4.5µM + BA 8.9µM	Anthers, ovaries	[79]
Long-term SE, storage	2,4 D 9 μM + IAA 6 μM + BA 4.4 μM + GA31.8 μM	In vitro leaves	[80]
Long-term SE	2,4-D 9 µM + BA 1 µM + IASP 17 µM then 2,4-D 2 µM or 2,4-D 2 µM + IASP 4 µM	Ovary	[81]
Long-term SE	2,4-D 2.25 μM + BA 18 μM	Anther/filament	[82]
Long-term SE	2,4-D 9 µM + BA 0.9 µM	Anthers	[83]
Medium influence, protocol development	2,4-D 2.5 µM + BA 0.8 µM	Anthers	[84]
Mutagenesis	2,4-D 9 µM + BA 4.4 µM then NAA 5.4 µM + BA 4.4 µM	Leaves	[85]
Physiology of SE	2,4-D 4.5 µM + BA 1.1 µM	Anthers	[86]
Physiology of SE	2,4-D 4.5 μM + BA 9 μM	Immature anthers	[25]
Physiology of precocious germination	2,4-D 1 μM +TDZ 4.5 μM	Filaments	[87]
Ploidy change	2,4-D 1 μM + TDZ 4.5 μM	Anther/filaments	[88]
Ploidy change	2,4-D 1 μM+ TDZ 4.5 μM	Anther/filaments	[34]
Protocol improvement	2,4-D 4.5 µM + BA 9 µM	Anther/filaments	[24]
Protocol improvement	2,4-D + TDZ several combinations	Anthers	[89]
Protocol improvement	2,4-D 4.5 µM + BA 1.1 µM	Anthers	[90]
Protocol improvement	2,4-D 5 µM + BA 0.9 µM	Anthers	[91]
Protocol improvement	2,4-D 5 µM + BA 1 µM	Anthers	[92]
Protocol improvement	2,4-D 20 µM + BA 9 µM	Anthers	[26]
Protocol improvement	2,4-D 5 µM + BA 1 µM	Anthers	[93]
Protocol improvement	2,4-D 5 µM + TDZ 0.2 µM; 2,4-D 5 µM + BA 0.4 µM; 2,4-D 2.5 µM + NOA 2.5 µM + 4-CPPU 5 µM	Anthers	[94]
Protocol improvement	2,4-D 4.5 μM + BA 4.4-μM	Anthers, gynoecia	[95]
Protocol improvement	2,4-D 9 µM + BA 4.4 µM	Anthers, ovary	[96]
Protocol improvement	2,4-D 4.5 µM + BA 8.9 µM	Anthers, ovary	[97]
Protocol improvement	2,4-D 1 µM + TDZ 1 µM	Filaments	[98]
Protocol improvement	2,4-D 4.52 µM + BA 4.4 µM; 2,4-D 4.52 µM + NOA 2.5 µM + 4-CPPU 5 µM	Ovary, immature anthers	[99]
Protocol improvement	2,4-D 4.5–9 µM + BA 4.5–9 µM	Ovary, immature anthers	[100]
Protocol improvement	NOA 2.5 µM + BA 5 µM + 2,4-D 2.5 µM	Immature leaves	[101]
Protocol improvement	2,4-D + BA+ NOA several combinations	Immature leaves, stamen	[102]
Protocol improvement	TDZ 0.90 μM	Immature seeds	[103]
Protocol improvement	NOA 5 μM + BA 4.5 μM + several aminoacids	In vitro leaves	[104]
Protocol improvement	2,4-D 5–10 µM + TDZ or 4-CPPU 5–10 µM	Leaves	[105]
Protocol improvement	NOA 20 µM + BA 40 µM or TDZ 4 µM	Leaves	[106]
Protocol improvement	2,4-D 9 µM + BA 9 µM	Leaves	[107]
Protocol improvement	2,4-D 0.45 µM + BA 4.5 µM	Leaves	[108]
Protocol improvement	IAA 5.7 µM or IBA 0.5 µM	Leaves, petioles	[22]
Protocol improvement	2,4-D 9 µM + BA 4.4 µM, then NAA 10.7 µM + BA 0.9 µM	Leaves, petioles	[109]
Protocol improvement	2,4-D 5 µM + BA 1 µM	Ovules	[110]
Protocol improvement	2,4-D 1 µM + TDZ 0.2 µM	Ovules	[23]
Protocol improvement	2,4-D 2 µM	PEMs from anthers and ovary	[111]
Protocol improvement	BA 2.2 µM	Petioles	[112]
Protocol improvement	NOA 20 µM + TDZ 4 µM	Protoplasts	[113]
Protocol improvement	NAA 10.7 µM + BA 2.2 µM	Protoplasts	[27]
Protocol improvement	2,4-D 9 µM + BA 4 µM	Seed integuments	[114]
Protocol improvement	2,4-D + BA several concentrations	Stem segment	[115]
Protocol improvement	2,4-D 4.5 µM + BA 9 µM	Whole flowers, anthers, ovary	[116]
Protocol improvement	2,4-D 4.5 µM + BA 9 µM	Whole flowers, anthers, ovary	[117]
Protocol improvement	NOA 5 µM + BA 0.9 µM	Zygotic embryos	[118]
Protocol improvement	NOA 5 µM + BA 0.9 µM	Zygotic embryos	[119]
Protocol improvement for hybrids	2,4-D 5 µM + BA 1 µM	Anthers	[120]
Protocol improvement for hybrids	2,4-D 5 µM + BA 1 µM	Anthers	[121]
Regeneration from protoplast	NAA 10 µM + BA 0.1 µM	Protoplast from embryogenic callus produced from stamen	[20]
Rejuvenation of cell cultures	2,4-D 4.5µM + BA 0.1 µM	Anthers	[122]
Sanitary improvement	2,4-D 9 μM + TDZ 10 μM	Anther/filament	[35]
Sanitary improvement	2,4-D 4.5 μM + BA 8.9 μM	Anther/filament, pistils	[123]
Sanitary improvement	TDZ 0.90 μM; TDZ 0.45 μM	Immature seeds	[103]
Sanitary improvement	TDZ 0.90 μM +2,4-D 4.5 µM	Anther/filament	[124]
Sanitary improvement	Review	Review	[125]
Sanitary improvement	2,4-D 4.5 μM+ BA 8.9 μM	Immature anthers, ovaries	[126]
SE for biodiversity conservation	NOA 5 μM+ BA 4.4 μM	Anthers, pistils	[127]
Synchronization of development	1 μM 2,4-D, 4.5 μM TDZ,	Anthers/filaments	[128]
Virus elimination	BA, NOA, 2,4-D several combinations and ratios	Anthers	[129]

Abbreviations: 2,4-D: 2,4-dichlorophenoxyacetic acid; BA: 6-benzyladenine; 4-CPPU: N-(2-chloro-4-pyridyl)-N′-phenylurea; IAA: indole-3-acetic acid; IBA: indole-3-butyric acid; IASP: indole-3-acetyl-L-aspartic acid; NAA: 2-naphtaleneacetic acid; NOA: 2-naphthoxyacetic acid; PEM: proembryogenic masses; PGRs: plant growth regulators; TDZ: N-(1,2,3-thidiazol-5-yl)-N′-phenylurea (thidiazuron).

**Figure 1 plants-13-03131-f001:**
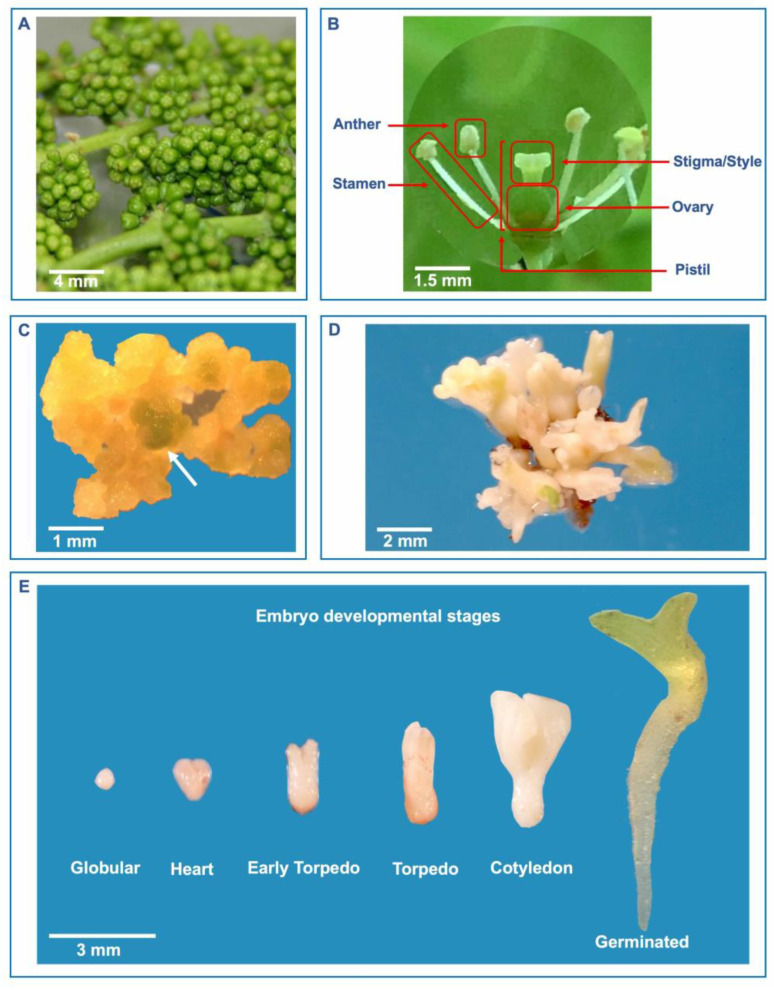
Somatic embryogenesis and plant regeneration from immature flower tissues in *Vitis vinifera*. (**A**) Immature flowers collected a few days before opening and stored at 4 °C. (**B**) Different floral tissues utilized to induce embryogenic cell lines (anthers, pistils, stigmas/styles, ovaries, and whole flowers). (**C**) Callus generated from a pistil (arrow) after 3 months of culture on embryogenic medium. (**D**) Somatic embryos regenerated after 4–6 months of culture initiation at the surface of explant-derived callus. (**E**) Different developmental stages of somatic embryos regenerated in vitro. The response to somatic embryo induction can also vary based on the organ/tissue types. Anthers, ovaries, leaves, petioles, tendrils, and nodal sections are the explants that are most frequently used for SE induction in grapevine [36]. A remarkable number of genotypes have been regenerated through anther culture [130] with a high success rate. Nevertheless, some authors report that, depending on the stage of growth of reproductive organs, the response can change. For instance, according to Vidal et al. [99], the regeneration from the ovaries was around two times greater than that from the anthers when ovaries were cultured in later stages of development. According to a recent study by San Pedro et al. [103], mature seeds can also be used as explants for SE induction by holding cut seeds for five months in media supplemented with TDZ. However, zygotic seed-derived somatic embryos are not useful for clonal propagation. Embryogenic callus induction has also been achieved using nodal segments, leaf discs [131], petioles, stem nodal explants [115,132], and whole flowers [66], even though they are less commonly used.

It is widely known that the developmental stage of the explant affects the effectiveness of SE induction. Additionally, exogenous application of the right auxin at right concentration and the duration of incubation also have considerable influence on the success of the protocol [99,133]. The first factor to consider while choosing anthers at a particular developmental stage is the size of the floral buds. It has been reported that buds 1.55 cm long on average have anthers enclosing uninucleate microspores, which are the most responsive to SE in *Vitis*. Moreover, in proportion to their stage of differentiation, explants’ ability to alter their evolutionary trajectory diminishes [134], and this seems to be the case both for carpels and for stamens [133]. Similar conclusions have been drawn by other authors [4,47,135]. Vidal et al. [136] showed that earlier flower developmental stages are more conducive for embryogenic culture induction from anthers, while later stages are more conducive for induction from ovaries. Three flower developmental stages were classified by Prado et al. [100]. R1 and R2 are equivalent to stages V and VI according to Gribaudo et al. [116], while R3 is the late binucleate microspore stage. The authors reported that two cultivars—‘Mencía’ and ‘Brancellao’—were best utilized at the R3 stage, whereas four cultivars—‘Albaríio’, ‘Treixadura’, ‘Torrontés’, and ‘Merenzao’—showed the best results at the R2 stage [100]. Similarly, two *V. vinifera* cultivars—‘Chardonnay’ and ‘Barbera’—responded better when anthers of early-stage microsporogenesis were cultured, whereas for the rootstock ‘110 Richter’ (*V. berlandieri* × *V. rupestris*), more embryogenic cultures could be established using explants in the later stages of maturation [116].

## 3. Stages and Synchronization of Somatic Embryo Production in Grapevine and Their Germination

The morphological and temporal development of somatic embryos proceed through a series of distinct stages, with globular, heart, torpedo, and cotyledon or plantlet stages for dicotyledons [130,137] and globular, elongated, scutellar, and coleoptilar stages for monocotyledons [138]. In grapevine, embryos go through distinct phases, such as globular, heart, torpedo, and early cotyledon stages, before germinating (Figure 1E) [51].

Globular embryos usually appear on the surface of embryogenic calli (Figure 1C,D) and have no vascular connections. The young embryo is circular or slightly oblong with small cells having thick walls and is in close contact with the callus from which it was generated [139]. When the embryo detaches from the callus, axial cells start to elongate, marking the beginning of the tissue differentiation process. The two apical meristems present by the end of the globular stage persist through the maturation step, during which embryos pass from the heart-shaped to the torpedo stage and cotyledons expand due to the deposition of storage materials [140,141]. Mature embryos, having accumulated enough storage materials, develop into normal plants, passing from the torpedo stage to the germinated embryo [142]. Grapevine somatic embryos show radicle growth, tannin accumulation in the central cylinder, and acquisition of an external suberin sheath [86,143].

Automation and scaling up are needed to improve cost-effectiveness in every process. To maximize the output of SE and lower unit costs through automation, it is imperative to synchronize the growth of somatic embryos in embryogenic suspension cultures. Synchronization of somatic embryo production is highly desirable for applications in micropropagation, genetic transformation, and gene expression studies related to SE.

Jayasankar et al. [144] compared somatic embryo development on solid media with embryos cultured in liquid media. On agar-based medium, somatic embryos had large cotyledons, a negligible or absent suspensor structure, and a relatively undeveloped concave shoot apical meristem, whereas those grown on liquid medium showed a distinct suspensor and a flat-to-convex shoot apical meristem enclosed in smaller cotyledons. Only the somatic embryos grown on solid media exhibited dormancy. The authors hypothesize that the presence of a persistent suspensor, typical of somatic embryos regenerated in liquid media, modulates development, ultimately resulting in rapid germination and a high plant-regeneration rate [144].

Jayasankar et al. [145] observed that embryogenic suspension cultures of *V. vinifera* ‘Thompson Seedless’ did not progress beyond the heart shape when the auxin 2,4-dichlorophenoxyacetic acid (2,4-D) was removed from the medium, while those of ‘Chardonnay’ developed fully. To synchronize the development of somatic embryos, they inoculated <960 µm fraction into media without auxin. Sieving of proembryogenic masses (PEMs) and subculture resulted in the synchronization of embryo development and reduced browning and abnormalities, such as fasciation or fusion, during differentiation. As a result, they achieved 60% conversion of SEs into plants. On the other hand, when 2-naphthoxyacetic acid (NAA) was used in liquid culture, PEMs of ‘Chardonnay’ clone 76 showed poor competence for further development when the auxin was removed [146]. However, growth and development could be stimulated by daily subcultures, and the authors attributed the arrest of development under standard subculture conditions to the accumulation of extracellular macromolecules with molecular weights > 10 kDa [146]. When embryogenic-competent ‘41B’ (*V. vinifera* cv. Chasselas × *V. berlandieri*) cultures were compared with ‘Chardonnay’ clone 76 that showed arrest at heart stage, the protein patterns in auxin-enriched culture media were practically identical. When the auxin was removed, extracellular proteins of 38, 51, and 62 kDa were over-accumulated in the ‘Chardonnay’ clone 76 cell culture compared to the 41B cell line, whereas 36 and 48 kDa proteins were excreted only by the 41B cell line. These differences were attributed to the differences in embryogenic competence in the two cell lines [147]. On the other hand, Zlenko et al. [148] successfully converted somatic embryos that had been grown on liquid induction media by subculturing them on liquid media that had either GA_3_ alone or a combination of BA and GA_3_ added. Solid media, either with or without BA, were successfully employed to promote plant development. Also, Vasanth and Vivier [59] and Wang et al. [149,150] used liquid media to produce synchronized somatic embryos for cryopreservation procedures.

Several factors influence the rate of conversion of somatic embryos, and abnormalities can be due to genetic or epigenetic changes in the DNA. Stress factors such as high and low temperatures, drought, salt, heavy metals, and usage of mutagenic chemicals and PGRs can result in modifications in DNA [151]. Among others, ABA metabolism has been reported as crucial in the maturation of grapevine somatic embryos [87]. Despite these differences, a significant benefit is the capacity to sustain embryogenic cultures over an extended period of time without losing their embryogenic potential, as demonstrated by numerous studies [69,79,81,83,105,152].

## 4. Factors Affecting Somatic Embryogenesis in Grapevine

SE in grapevine is influenced by several factors, both internal and external, which makes the application of the technique challenging. Embryogenesis processes are notably impacted by the selection of the appropriate explant, medium, PGRs, genotype, carbohydrate, and gelling agent, among other factors, including light regime, temperature, and humidity [47,100,153].

### 4.1. Genetic Control

Although in many species all cultivars have comparable levels of embryogenic potential, in grapevine, SE competence is strongly genotype-dependent. The embryogenic potential of cultivars and even clones within a cultivar can vary considerably. For example, in a comparison of eight grapevine cultivars for embryogenic potency, about a 50-fold difference was observed [47]. Although multiple methods have been published, for certain cultivars the technique still needs further improvement [58,154]. Several authors have highlighted the different responses to SE of grapevine according to genotype [47,58,94,116,135,152]. It is commonly recognized that genotypes used as rootstocks have a stronger capacity for regeneration through both organogenesis and SE than *V. vinifera* hybrids and cultivars. For example, among three different Italian *V. vinifera* L. cultivars and four hybrid rootstocks, SE efficiency was higher for rootstocks irrespective of the medium and explant used [76].

SE in grapevine is governed by complex gene signaling networks involving transcriptional regulation, auxin and protein signaling, and extracellular matrix interactions. Auxin plays a pivotal role, as auxin gradients influence cell fate and differentiation. In grapevine, the genes *VvPIN* and *VvAUX1/LAX* are critical for creating these auxin gradients. *VvPIN* genes encode auxin efflux carriers that facilitate the directional transport of auxin out of cells, while *VvAUX1* and *LAX* genes encode auxin influx carriers that mediate the uptake of auxin into cells. The interplay between these transporters is essential for maintaining auxin homeostasis and creating localized auxin maxima that drive development and organ formation [28,32,155]. The model of the PIN efflux system-dependent auxin gradients in the control of growth and patterning in zygotic embryogenesis, reviewed by Elhiti and Stasolla [156], applies to SE as well. Exogenous application of auxin, a critical requirement for SE, triggers a general reprogramming of somatic cell transcriptomes and modulates the expression of many SE-associated transcription factors. High concentrations of auxin in cultures result in a significant increase in the size of nuclei, suggesting a reorganization of the chromatin [157]. This stress leads to a modification of chromatin state, which in turn results in the activation of transcription factors, such as *WUS*, *LEC* genes, and *BBM*, that are specific to embryogenic programs [4,158]. These results suggest that PGRs, especially auxin, trigger a general reprogramming of gene expression through chromatin modifications and activation of specific transcription factors.

The genes expressed during grapevine SEs have been studied, particularly the *Somatic Embryogenesis Receptor Kinase* (*SERK*) and *Leafy Cotyledon* (*LEC* and *L1L*) genes [159]. These genes play important roles in SE in various plant species. The expression of *VvSERK1*, *VvSERK2*, *VvSERK3*, and *VvL1L* genes has been analyzed during SE in grapevine, and the results showed that these genes are involved in the regulation of SE, with expression of *VvSERK2* being relatively stable during in vitro culture and *VvSERK1*, *VvSERK3*, and *VvL1L* being expressed more 4 to 6 weeks after transfer of the calli onto embryo induction medium before the appearance of embryos on calli. After 8 weeks in embryo induction medium, *VvSERK1* was expressed in the calli and *VvSERK3* in the embryos. Expression of *VvL1L* was low at this time [159]. Thus, the differential expression of key genes, such as *SERK* and *LEC1-like*, is crucial for the embryogenic process, as these genes play pivotal roles in promoting somatic embryo formation through stress and developmental signaling pathways [160]. Additionally, lipid-transfer proteins (LTPs) secreted during SE function as extracellular signaling molecules that are vital for proper cell-to-cell communication and membrane dynamics [161]. Overexpression of the *VvLTP1* gene, however, disrupts normal embryo development, indicating that precise regulation of *LTPs* is essential for maintaining the balance of signaling necessary for embryogenesis [162]. Proteolytic regulation, facilitated by extracellular proteins and protease inhibitors, further influences the embryogenic process by modulating the extracellular matrix and thereby affecting gene signaling [163]. Moreover, distinct extracellular protein patterns observed in different embryogenic states underline the significance of the extracellular environment in shaping gene expression and subsequent developmental outcomes [147]. These studies collectively highlight the intricate interplay between intracellular signaling and extracellular factors in regulating somatic embryo formation in grapevine.

### 4.2. External Factors Controlling Somatic Embryogenesis

Explant type as a major factor determining the success of SE in grapevine has already been discussed in detail in Section 2. In this section, we discuss other external factors that can be controlled to optimize the process. The composition of the culture medium has a significant role in the success of plant regeneration because it supplies essential nutrients for the growth of explants at various developmental stages. Usually, the media adopted to induce SE in vitro are based on MS [164] or NN [165] salts. However, media formulations differ among different laboratories, and many different types of basal culture media have been tested, such as LS [166], WPM [167], C2D [168], and DKW [169]. Focused research on the effects of micro- and macronutrients is rarely reported. Nevertheless, it is known that ammonium promotes SE induction in some media [84]. The only carbohydrate supply used for embryogenic culture, SE induction, and development is sucrose at 10–180 gL^−1^, and it is most widely used at 30–60 gL^−1^. Sucrose plays an important role also as osmoticum for SE germination and plant regeneration because dehydration of grapevine SE may increase plant development [170]. To improve the grapevine regeneration process, some other protocols suggest supplementing media with amino acids such as glycine, phenylalanine, and L-glutamine [104,171].

The production of an embryogenic callus has been shown to be significantly influenced also by the type and concentration of PGRs (Table 1), such as 2,4-D, N-(2-chloro-4-pyridyl)-N′-phenylurea (4-CPPU), benzyladenine (BA), gibberellic acid (GA3), IAA, indole-3-butyric acid (IBA), NAA, NOA, and TDZ [172]. Usually, a combination of auxins, mainly 2,4-D or NOA, and CKs, mainly BA, added at different concentrations based on the type of explant and genotype is used to initiate embryogenic cultures. Some combinations that have been used to induce SE successfully include IAA combined with GA3 for fertilized ovules and urea derivatives like TDZ or 4-CPPU in combination with auxins in the induction phase in anther culture [67]. The continuous presence of PGRs is not suitable for somatic embryo development. For this reason, after an embryogenic callus has been induced, in some cases auxins are removed, decreased, or substituted with other PGRs. A comprehensive list of PGR combinations used for SE induction in grapevine can be found in Carimi et al. [173] and in Table 1.

Exogenous application of auxin results in a modification in endogenous auxin signaling. For example, in *Coffea canephora*, there is an increase in the content of endogenous IAA and in the expression of the genes that code for the enzyme tryptophan aminotransferase, which are involved in the biosynthesis of IAA [174]. While facilitating SE induction, exogenously applied synthetic PGRs can also disrupt the endogenous auxin balance and its transport, which can result in abnormalities in somatic embryos. This problem has been highlighted in a recent review by Garcia et al. [151] focusing on 2,4-D-mediated abnormalities. Using *V. vinifera* cv. ‘Chardonnay’ cotyledonary embryos with distinct morphologies as model systems, Ya et al. [175] demonstrated that the cellular concentrations of IAA and ABA were significantly higher in normal cotyledonary embryos compared to vitrified, fused, or elongated cotyledonary embryos. Comparative transcriptome analysis also revealed significant differences in gene expression of the hormone signaling pathways in normal and abnormal cotyledonary embryos of grapevine, providing further evidence of the importance of regulation of endogenous auxin and ABA for the production of healthy and viable somatic embryos through proper regulation of exogenously applied auxin.

During the early stages of embryogenesis, endogenous auxin levels rise significantly—a phenomenon that is associated with the activation of stress signaling pathways and alterations in chromatin structure. Several studies have demonstrated that exogenously applied auxins elevate the levels of endogenous IAA in explants undergoing SE. While the induction of embryo identity in somatic explants does not depend on endogenous auxin biosynthesis, maintaining embryo identity requires an increase in endogenous auxin levels. This elevation, along with proper auxin transport, is crucial for promoting the differentiation of embryonic cells into histo-differentiated somatic embryos [176]. It has also been reported that 2,4-D promotes the production of IAA-binding proteins, enhancing the sensitivity of cells to IAA and rendering them competent for embryogenesis [177]. Thus, the accumulation of endogenous auxin is crucial for altering cell fate and laying the foundation for embryogenic processes [178].

Also, physical culture conditions could significantly influence SE induction and regeneration frequency. Some authors indicate that a two-week culture period in the dark is useful for improving regeneration percentages [179]. Additional treatments aimed at enhancing regeneration efficiency include the use of activated charcoal [180], pretreatment with chilling [181], cotyledon removal [69,182], and adjusting pH levels [183]. The kind of culture—liquid or solid—also can affect the outcome; in the initial induction phase, liquid cultures are preferable, but organized embryogenic calli develop more readily on solid media [184].

Liquid suspension cultures are generally more efficient than solid media because plant cells are better exposed to medium components and the uptake and consumption of nutrients are faster. Liquid cultures allow for a higher cell growth rate and are more effective in regenerating somatic embryos. However, they are considered more complex when compared to cell cultures on solid media because they require shorter subculture intervals (tend to senescence earlier) and are more susceptible to bacterial and fungal contamination. Liquid suspension cultures of grapevine are generally started from about 200 to 400 mg of PEM incubated in 50 mL of liquid culture medium (Figure 2A,B), and after about two months new somatic embryos are regenerated (Figure 2C). The embryos are separated from the undifferentiated cells by filtration of culture using a nylon mesh filter and then incubated for about 30 days on liquid medium without PGRs (Figure 2D) and subsequently placed on solidified medium to germinate (Figure 2E). Within 4–6 months from the culture initiation of PEMs it is possible to have acclimatized plants (Figure 2F).

## 5. Applications of Somatic Embryogenesis and Embryogenic Cultures in *Vitis*

SE possesses a wide array of potential applications in micropropagation, germplasm conservation, and sanitary and genetic improvement, including the most modern genetic engineering techniques (Table 1). Herein, we summarize the state of the art and the impact that SE may have on grapevine propagation, conservation, and crop improvement.

### 5.1. Somatic Embryogenesis for Germplasm Management

As a clonally propagated heterozygous species, it is not possible to conserve grapevine clones used in winemaking through seed banking. This same principle applies to rootstocks, as these have also been selected for performance. For example, in France there are 15 certified clones of the Richter 100 rootstock [185]. Long-term conservation of vegetative tissue is not as easy as that of seeds, and therefore grapevine germplasm is maintained in field collections in many countries [186,187,188], leading to the erosion of valuable germplasm resources [189,190,191]. As an alternative, grapevine clones are maintained in slow-growth media in tissue culture [192,193,194], including rootstocks [195] and hybrid material used in breeding [196]. With SE as a better option due to its easier handling and storage, prolonging the lifespan of somatic embryos is important from a conservation perspective. Long-term storage of in vitro cultures has been facilitated by increasing sucrose levels to 7.5%, eliminating PGRs from media, and storage under low temperatures of up to 2 °C [197,198,199], but Pedro et al. [193] reported slowing of growth rates by halving the sucrose concentration in media. A gradual decrease in temperature over a few days facilitates better survival compared to transferring material to a low temperature abruptly [199]. Addition of 1.5 g L^−1^ of sorbitol or mannitol also reduced the growth of cultures [200]. Hassan et al. [201] reported prolonging storage of grapevine in vitro cultures by reducing sucrose concentration from the standard 30 g L^−1^ to 20 g L^−1^ and including 10 g L^−1^ sorbitol. Depending on the cultivar, they successfully used up to 50 g L^−1^ sorbitol, enabling maintenance for one year without subculture.

Compared to shoot cultures, longer periods of storage can be achieved using somatic embryos as the conservation propagules. Jayasankar et al. [184] demonstrated that drying suspension culture-derived somatic embryos to 25% of their initial weight over a laminar hood and storage in tightly sealed Petri dishes at 4 °C can extend the storage time to 42 months. In contrast, in another experiment, plant recovery was only 32% after 21 days of dehydration of somatic embryos under 70% humidity [202]. As SE is so far the only pathway for genetic transformation in grapevine [74], our ability to maintain the long-term viability of somatic embryos is also important for grapevine transformation providing an uninterrupted supply of plant material. The difficulty of storage of somatic embryos, unlike true seeds, is because somatic embryos lack desiccation tolerance. By mimicking the process of acquisition of desiccation tolerance during sexual seed development, Senaratne et al. [203] were able to produce alfalfa somatic embryos which can be dried to 8–15% moisture without losing viability. For this, they incorporated ABA in media during the cotyledonary stage of development in a synchronized system. Hence, to achieve the goal of extending the shelf-life of grapevine somatic embryos, research on the synchronization of the SE process should proceed alongside research on the acquisition of desiccation tolerance. Faure et al. [204] showed that *V. vinifera* ‘Grenache noir’ does not have a peak of ABA mid-embryogenesis. This cultivar shows precocious germination. The authors hypothesized that the switch from mid- to late embryogenesis is not triggered by low endogenous levels of ABA and suggested exogenous application of ABA to prevent precocious germination and trigger late embryogenesis. Later, Goebel-Tourand et al. [205] showed that exogenous application of ABA can improve the maturation process of grapevine somatic embryos, reducing precocious germination. Gene expression studies in maturing grapevine somatic embryos have demonstrated the involvement of ABA biosynthesis in precocious germination vs. proper maturation [87]. Although the involvement of ABA metabolism in the maturation of grapevine somatic embryos has been demonstrated [87], research is lacking on the use of various stress factors to stimulate endogenous ABA biosynthesis during the process. Further research on these aspects would facilitate improvement of the maturation of somatic embryos, making them ideal propagules for storage of clonal collections.

Cryopreservation is considered the best method for storing germplasm efficiently and safely long-term, particularly for the conservation of vegetatively propagated species [206,207]. Two decades ago, two-step cooling procedures were successfully used to cryopreserve grape embryogenic cell suspensions, using both encapsulation vitrification [150,208] and encapsulation dehydration [60,149,208,209] methods. For example, using a two-step freezing procedure (−0.5 °C min^−1^ to −40 °C followed by immersion in liquid nitrogen) for embryogenic cells pretreated for 1 h with 0.25 M maltose and 5% dimethyl sulfoxide at 0 °C, Dussert et al. [210] achieved a 60% survival rate of cryopreserved somatic embryos. Different modifications for encapsulation vitrification and encapsulation dehydration were also studied during this period. Gonzalez-Benito et al. [211] achieved 45–60% viability by cryopreserving embryogenic cells encapsulated in alginate beads and cultured in liquid media with increasing sucrose concentrations (0.25, 0.5, 0.75, and 1 M, one day for each step) followed by desiccation in the air flow of a laminar flow cabinet for 2–4 h. A similar procedure with an additional two days of incubation in 1 M sucrose, resulting in desiccation of cells to 20.6% moisture, resulted in about 78% viability [149]. Solid media were better than liquid media for post-thaw regeneration [149]. In encapsulation vitrification, the dehydration of cells is achieved using a vitrification solution. After a preculture step in 0.75 M sucrose, Wang et al. [150] encapsulated embryogenic cells and used Plant Vitrification Solution 2 (PVS2) [212] to dehydrate alginate beads. They achieved 42–76% regrowth when the beads were treated with PVS2 solution for 270 min at 0 °C.

With the development of vitrification methods, particularly droplet and cryoplate methods, cryopreservation of cells and other tissues has become easier and applicable to many species, including grapevine [213,214,215]. Droplet vitrification is a simpler method applicable to a wide range of species. Recent studies with several grapevine genotypes revealed that somatic embryos are more amenable to droplet vitrification cryopreservation than shoot tips and axillary buds from in vitro grown plantlets [213]. Furthermore, Carimi et al. [127] were able to induce somatic embryos from pistils and anthers of the progenitor of cultivated grapevine, *V. vinifera* ssp. *Sylvestris*, and used axillary buds from germinated somatic embryos for cryopreservation by droplet vitrification with a success rate of up to 44%. As cryoplate-based techniques use alginate beads, plant regeneration takes longer than shoot tips cryopreserved using droplet vitrification [216]. Widely observed differential responses among genotypes [47,84,94,217] and interactions of genotypes with explants and media [58,78,84] can pose challenges in the use of somatic embryos as explant sources for the conservation of large collections. Nevertheless, according to some focused studies, medium optimization for multiple genotypes is possible [24]. Another barrier to the use of somatic embryos as propagules for the conservation of vegetatively propagated species such as grapevine is the possibility of separation of chimeras in cultivars that have cell layers of different genetic backgrounds [56,218,219] as well as somaclonal variation [36,220,221]. However, some studies have shown the genetic integrity of somatic embryo-derived plants using molecular markers [47,222].

### 5.2. Somatic Embryogenesis as a Tool for Sanitation

The vegetative propagation and exchange of budwood among grapevine-growing regions and countries contribute to the spread of grapevine pathogens. The perennial life cycle results in the spread of these diseases within vineyards. With 86 different virus species known to infect *Vitis* spp., grapevines host the most viruses among cultivated species [223]. Among these, fanleaf and leafroll diseases are the most damaging and widespread [224]. It has been estimated that fan leaf disease caused by a nepovirus (Grapevine fanleaf virus; *GFLV*) causes economic losses amounting to USD 16,600 per ha, and in France, where about two-thirds of vineyards are affected, the economic impact is at least USD 1.5 billion per year [224]. Among the five serologically distinct Closteroviridae viruses known to cause leafroll disease, Grapevine leafroll-associated virus 3 (GLRaV 3) is the most devastating. Leafroll disease is estimated to cause losses of USD 25,000 to USD 226,000 per ha over a 25-year vineyard lifespan depending on the location and cultivar [224]. Therefore, the establishment of vineyards free of damaging grapevine viruses is an important control measure. To this end, many countries have implemented sanitary selection programs and certification of clonal stocks. However, once a stock is infected, it is important to have robust methods to eliminate the infecting viruses.

Several methods have been applied to eliminate viruses from infected grapevine clones. Traditionally heat therapy has been used to reduce viral loads, but some viruses are heat-stable [225]. Although heat therapy is useful in reducing the incidence of a disease, when used alone it is often not useful for clean stock programs. Therefore, a combination of heat therapy with microshoot culture is often used in grapevine virus eradication programs [225], particularly for nepoviruses such as GFLV that readily infect even the meristem [38]. A grapevine microshoot consists of the meristem and 2–3 leaf primordia and is less than 0.5 mm [225]. However, the persistence of virus particles (e.g., GLRaV 3) in lower parts of the apical dome in 0.5 mm microshoots [226], some viruses infecting even the apical dome [227,228], possible cross-contamination during excision combined with difficulty in precise excision of microshoots [229], and poor regeneration of microshoots in some cultivars [230] have led to the emergence of more precise in vitro-based methods for virus eradication in horticultural species, including grapevine. These include cryotherapy [226,229,231,232], electrotherapy [231], and in vitro chemo- [233,234] and thermotherapy [235,236] applied separately or in combination.

Similar to other in vitro-based therapies, regeneration from somatic embryos has also become an important tool to eliminate viruses from infected grapevine clones. The high efficiency of virus eradication through SE can be explained by the origination from a single cell or a few organized embryogenic cells that lack vascular connections to the maternal tissue [144,237]. Secondary somatic embryos are generally attached to the root primordia of the parent embryo by a suspensor-like structure, again without any vascular connection to maternal tissue or to one another [238]. Nevertheless, Goussard et al. [239] were able to demonstrate only the elimination of leafroll-associated viruses but not *GFLV* in somatic embryo-derived plantlets originally initiated from ovaries. When somatic embryos were produced at 35 °C (thermotherapy) in the dark, Goussard and Wiid [39] were able to remove *GFLV* in addition to leafroll viruses. Plantlets derived from somatic embryos produced at 25 °C were still infected with *GFLV* [39]. Nepoviruses such as *GFLV* can readily invade plant meristems [38]. Using three GFLV-infected Italian cultivars, Gambino et al. [38] demonstrated the presence of the virus in all tested anthers and ovaries and the calli derived from both these explants. Nevertheless, only a few somatic embryos of one cultivar and only one out of sixty-three plants tested during micropropagation of somatic embryo-derived plantlets tested positive, while all the tested plants after one or two dormancy periods in the greenhouse were negative for the virus [38]. Similar results were reported for three *V. vinifera* cultivars infected with GLRaV-1, GLRaV-3, GVA, and GRSPaV. Four months after culture initiation, higher infection rates were reported in ovary cultures compared to anther cultures, but after eight months of culture, none of the tested cultures were positive for any of the viruses, with similar results for individual somatic embryos tested. All regenerated plantlets (12 months after culture initiation) and greenhouse plants (24 months after culture initiation) were free of the viruses [240]. High-throughput sequencing and RT-PCR have been used to compare the efficiency of SE and meristem culture for the elimination of several viruses in grapevine [124]. The results showed that SE using anthers with filaments as explants was effective for eliminating various grapevine viruses, including grapevine rupestris vein feathering virus (GRVFV), grapevine Syrah virus 1 (GSyV-1), grapevine virus T (GVT), and grapevine Pinot gris virus (GPGV) [124]. Rapidly proliferating cells and embryoids originating from these may escape infection [240], or it is possible that an embryogenic callus may originate from virus-free cells of the explant [238].

In conclusion, plant regeneration through SE from different explants of floral origin can be used to establish healthy grapevine stocks free from many grapevine viruses. Sanitation through SE is technically more difficult and time-consuming than traditional sanitation protocols [152]. Nevertheless, this technique is highly successful in grapevine. In the case of chimeric cultivars, virus elimination may be achieved either by traditional meristem tip culture and thermotherapy or by cryotherapy, sometimes requiring a combination of therapies [226,229,231,241].

### 5.3. Induced Mutagenesis for Grapevine Improvement

Many horticultural species, including grapevine, are maintained through vegetative propagation over multiple cycles. While this practice helps preserve superior agronomic traits in cultivars, the accumulation of somatic mutations results in phenotypic diversity. Mutants can be selected and propagated as new clones of the mother variety. This diversity in traditional cultivars is the basis for the selection of improved clones without losing varietal identity for the very traditional wine industry. A good example of such selection is the grape cultivar ‘Benitaka’ (red berries) that was selected from ‘Italia’ (green berries). The cv. ‘Brazil’ (black berries) was then selected from ‘Benitaka’ [242]. A sequence analysis of the promoter region and coding sequence of *VvmybA1* revealed a base substitution between ‘Benitaka’ and ‘Brazil’ in the promoter region and a deletion of a large DNA fragment in the promoter region of ‘Italia’. Anthocyanin contents and expression of the *VvmybA1* and *UFGT* genes in ‘Brazil’ were higher than in ‘Benitaka’ and barely detectable in ‘Italia’ [242]. Economically important clones of ‘Pinot noir’, ‘Cabernet sauvignon’, and ‘Chardonnay’ have been the result of clonal selection. Pinot is one of the oldest grape cultivars and is a noble cultivar used in many countries in different continents, including French Champagne and Bourgogne wines [243,244]. It displays extensive clonal diversity, and in France alone 64 different Pinot clones have been certified and marketed. Furthermore, approximately 95% of grapevine plants produced in French nurseries originate from clonal selection [243]. While naturally occurring mutations can produce agronomically valuable clones for selection, this process is slow [245] and not ideal for a breeding program.

Induced mutagenesis can increase the frequency of mutations in genomes [245,246]. Although more than 3300 induced mutants have been registered and published [245], the number of mutant cultivars in horticultural species is very limited [245,246]. Unlike in seed-propagated species, mutant selection in vegetatively propagated crops is not straightforward. Using either ionizing radiation or mutagenic chemicals, it is possible to induce mutations in planting material, including tissue-cultured plantlets. However, in vegetatively propagated plants, following mutagen treatment, several cycles of propagation are needed to obtain homo-histonts or to ‘dissolve’ chimeras and to obtain ‘solid’ mutants [247]. This is because the meristem is multicellular and the cell with the desired mutation produces a sector, resulting in a chimera. The in vitro subculture of mutagen-treated material through several generations can be achieved more rapidly than by grafting or rooting of cuttings in classical vegetative propagation of grapevine. Even then, the resulting mutant is often a sectorial chimera [246,248,249]. Several researchers have used mutagen treatment of in vitro-cultured shoots for mutation induction, followed by several subcultures to remove the chimeras. Khawale et al. [250] used two nodal microcuttings of *V. vinifera* ‘Pusa Seedless’ for mutation induction in ethyl methanesulfonate (EMS) and ethidium bromide (EB)-supplemented culture media (ten concentrations from 0.01 to 0.1%). Based on in vitro survival of microcuttings and their subsequent in vitro growth response, the LD_50_ value for EMS was recorded as 0.04%, and for EB it was 0.06%. Randomly Amplified Polymorphic DNA (RAPD) markers were used to detect mutant plants after three subcultures. Seven out of thirty of the RAPD primers used showed polymorphisms in the mutant population [250]. Munir et al. [251] also used RAPD analysis to identify mutants after irradiation of cultures from three cultivars with gamma rays and reported high yield of mutants, based on polymorphisms for some of the RAPD primers used.

Use of SE is the solution to chimerism in mutation breeding, as somatic embryos arise from single cells. The existence of growth centers comprising 5–50 cells in embryogenic calli of grapevine was first reported by Krul and Worley [21]. Subsequent anatomical observations using scanning electron microscopy also failed to confirm the single-cell origin of grapevine somatic embryos due to the technical difficulties involved in observing the sequential development of single living embryogenic cells [237,238]. Gambino et al. [240] observed the differentiation and development of somatic embryos from fast-growing calli. However, Faure et al. [86] were the first to report the single-cell origin of grapevine somatic embryos, and this has been demonstrated in other species as well [252,253,254]. Thus, the SE-based regeneration approach has great potential for isolating ‘solid’ mutants in vegetatively propagated species [247,255,256,257,258]; however, only a few researchers have used embryogenic cultures of grapevine for mutation induction. The low rate of somatic embryo induction in many cultivars could be the reason for this [135,205]. Kuksova et al. [85] tested the effect of five doses of gamma rays (5–500 Gy) and exposure to 0.025 mM colchicine over 3 days on embryogenic cultures of *V. vinifera* ‘Podarok Magaracha’. They observed polyploidization with gamma rays (in 5–100 Gy treatments) but not with colchicine. Except for polyploids, only chlorophyll mutants were reported. The authors emphasized the value of the use of embryogenic cultures in mutagenesis, as none of the polyploids displayed chimerism for the chromosome numbers [85]. Yang et al. [259] used colchicine treatment of globular-stage somatic embryos derived from immature zygotic embryos of diploid *V. vinifera* ‘Sinsaut’. They were able to produce tetraploids when the cultures were treated with 20 mg L^−1^ colchicine for 1–3 days, with 1-day treatment producing the highest frequency of 4% tetraploids among regenerated somatic embryos. They also reported the uniformity of tetraploidy in the individual plants in repeated tests, confirming that SE-based mutagenesis can produce chimera-free mutants [259]. Polyploidization in grapevine may allow a greater fruit size and a delay in ripening time. Capriotti et al. [260] treated 2 mm slices of embryogenic masses of *V. vinifera* ‘Chardonnay’, ‘Melot’, and ‘Pinot Grigio’ with 0.05, 0.25, and 0.5% EMS solution for 3 h and 0.03 and 0.04% sodium azide for 4 h and regenerated over 1400 plants which were screened for a natural infection of powdery mildew (*Erisiphe necator*). They identified 5 Pinot Grigio, 81 Merlot, and 59 Chardonnay putative mutants showing low infection [260]. Pathirana and Carimi [58] optimized the EMS treatment of *V. vinifera ‘*Chardonnay’, ‘Sauvignon blanc’, and ‘Riesling’ for mutation induction and reported that treating somatic embryos with 0.1% EMS solution for one hour resulted in 50% survival, which they considered optimal for mutation induction experiments.

In Figure 3, we illustrate a scheme for mutation induction and selection using embryogenic cultures of grapevine that is applicable to any other crop. We suggest optimizing treatment with mutagens using growth reduction curves, as demonstrated in Figure 3 and in Pathirana and Carimi [58]. It is recommended that a mutant dose resulting in 50% growth reduction be used for inducing mutations in large populations of embryogenic cultures [246,249]. The regenerated embryos after mutagen treatment can be challenged in vitro for many agronomic traits, such as toxic chemicals, salinity, pH, drought, viruses, etc. [36,246,249,255], or they can be tested under greenhouse or field conditions (Figure 3).

Another approach for grapevine improvement would be the generation and screening of mutant populations developed through transposon activation in embryogenic tissues by exposure to stress treatments. Movement and insertion of transposons is an important source of variation and evolution in the plant kingdom [261]. Color variation in maize kernels due to transposon insertion [262,263] is a classic example. In grapevine, red berry variants often encountered as mutants in white berry cultivars have been shown to be the result of recombination between long terminal repeats of the Gret1 retrotransposon present in a homozygous state at the promoter of *VvMybA1* in white grapevine cultivars [264,265]. Furthermore, new-generation sequencing of phenotypically different ‘Pinot noir’ clones has revealed that insertion polymorphism generated by mobile elements displayed the highest number of mutational events with respect to clonal variation [243]. The publication of the complete grapevine genome has provided further evidence that mobile elements, in particular Class II elements, have contributed to the genomic variability of *V. vinifera* [266], and of the repetitive sequences representing 66.47% of the genome, the largest portion comprised transposable elements (63.90%) [267]. With several research groups reporting induction of grapevine secondary somatic embryos [47,51,96,115] and their cryopreservation [59,60,149,150,208,210,211], it is now possible to use RNAseq tagging and recover mutant embryos with the same mutation. This proposed scheme is presented in Figure 4.

### 5.4. Genetic Engineering

Somatic embryogenesis is a valuable biotechnological tool that allows for the genetic manipulation of clonally propagated species, such as grapevines [135,172,173,245,258,268]. Various grapevine varieties have been successfully genetically transformed using embryogenic cultures to produce highly regenerative target material [72,82,108,135,172,268,269]. The grapevine genetic background influences SE, which is the most commonly used regeneration method in genetic engineering protocols for this crop [135,268] and also provides a dependable method for clonal propagation, genetic enhancement, and functional genomic research following transformation [46,70,135,159,173,269,270].

Genetic engineering in grapevines can improve various traits relating to grapevine cultivation. It can lead to the development of stress-tolerant and disease-resistant varieties with increased productivity, efficiency, sustainability, and environmental adaptation [46,82,136,172,269,271,272,273,274,275]. However, the successful commercialization of genetically improved grapevine varieties faces several challenges. These include scientific, legal, and regulatory issues; intellectual property and patenting concerns; political and economic factors; and negative public perception of genetically modified products [173,276,277]. Overcoming these hurdles is crucial for the implementation and widespread adoption of genetically improved grapevine varieties.

Research has also been conducted on the characterization of tumorigenic strains of Agrobacterium spp. isolated from grapevine tumors [278]. These strains, including *Agrobacterium vitis*, *Agrobacterium tumefaciens*, and *Agrobacterium rhizogenes*, were found to be tumorigenic in grapevines and exhibited different pathogenicity levels in other hosts [278]. The study also identified chromosomal and Ti plasmid genes that can be targeted for PCR amplifications to detect these Agrobacterium species in grapevine [278]. Biological control of crown gall, a disease caused by *A. tumefaciens*, has been investigated in grapevine using nonpathogenic strains of *Rhizobium vitis* [279]. These nonpathogenic strains, such as ARK-1, have been shown to reduce the incidence of crown gall in grapevine plants [279]. The use of biological control agents like ARK-1 can provide an alternative to chemical treatments for managing crown gall disease in grapevine. Anthocyanin acyltransferases play a crucial role in the production of acylated anthocyanins in grape skins [280]. The regulation of these enzymes by transcription factors like *VvMYBA* can influence the composition of anthocyanins in grapes [280]. Understanding the biosynthesis and regulation of these compounds is important for determining the aroma profiles and quality of grapes and wines.

Genetic engineering techniques, such as Zinc Finger Nuclease (ZFN), Transcription Activator-Like Effector Nuclease (TALEN), and CRISPR/Cas, have been employed in *V. vinifera* research to improve traits such as disease resistance and sugar accumulation [43,44,272,281,282]. These techniques offer targeted genome editing capabilities, allowing for precise modifications to the grapevine genome. The CRISPR/Cas9 system has been used in grapevine to edit specific genes of interest. Wang et al. [282] performed whole-genome sequencing of Cas9-edited grapevine plants and identified rare off-target mutations. Wang et al. [282] and Ren et al. [44] optimized the CRISPR/Cas9 system in grapevine by using grape promoters, which significantly increased the editing efficiency. Wan et al. [46] reported the use of CRISPR/Cas9-mediated mutagenesis to enhance resistance to powdery mildew in *VvMLO3*-edited lines of the ‘Thompson Seedless’ cultivar, whereas *VvPR4b* knockout lines had increased susceptibility to *Plasmopara. viticola* causing downy mildew in the same cultivar [42]. In another study, knocking out one allele of *VvbZIP36* in grapevine with CRISPR/Cas9 promoted anthocyanin accumulation [45]. Using a CRISPR/Cas9 editing vector construction with the peroxide sensor HyPer as a reporter, Fizikova et al. [41] were able to select *MLO7* knockout mutants in *V. vinifera* ‘Merlot’.

In addition to CRISPR/Cas9, other genetic engineering techniques have been explored in grapevine. For example, Vidal et al. [283] demonstrated high-efficiency biolistic co-transformation and regeneration of grapevine plants containing antimicrobial peptide genes. When regenerated and acclimated plants were challenged in the greenhouse with either *A. vitis* strains (bacterial crown gall pathogen) or *Uncinula necator* (powdery mildew pathogen) for evaluation of disease resistance, a total of six mag2 (natural magainin-2) and five MSI99 (a synthetic derivative) lines expressing the antimicrobial genes exhibited significant reductions in crown gall symptoms as compared to nontransformed controls. However, only two mag2 lines showed measurable symptom reductions in response to *U. necator*, but not strong resistance. Their results suggest that the expression of magainin-type genes in grapevines may be more effective against bacteria than fungi [136].

Dhekney et al. [274] used cisgenic engineering to develop grapevines with improved fungal disease resistance by isolating and modifying the *V. vinifera* thaumatin-like protein gene. Bosco et al. [273] investigated the correlation between the expression of disease resistance in genetically modified grapevines and the contents of viral sequences in T-DNA and global genome methylation. These genetic engineering techniques offer potential solutions for improving disease resistance in grapevine. Traditional breeding methods have limitations in identifying *Vitis* species with virus resistance, making genetic engineering an attractive alternative [281,284]. By targeting specific susceptibility genes, genome editing technologies like CRISPR/Cas9 can decrease susceptibility to fungal and oomycete diseases in grapevine [272]. Furthermore, the introduction of genes with antimicrobial activity from other plants or microorganisms has been used to enhance resistance to fungal and bacterial diseases in grapevine [273]. He et al. [275] isolated a gene encoding a pathogenesis-related thaumatin-like protein from a clone of downy mildew-resistant *V. amurensis* and transformed it into SE calli of *V. vinifera* ‘Thompson Seedless’ via *A. tumefaciens*. The transgenic grapevines exhibited improved resistance against downy mildew, with significant inhibition of hyphae growth and asexual reproduction of the pathogen [275].

Transgenic grapevines have been the subject of several studies exploring different aspects of grapevine biology, biochemistry, and genetics. Rinaldo et al. [280] investigated the role of a grapevine anthocyanin acyltransferase gene, *VvMYBA*, in the production of acylated anthocyanins in grape skins. They ectopically expressed the *VlMYBA1* gene from *V. labruscana* in grapevine hairy root tissue and analyzed gene expression changes in the transcriptomes of these roots. They found that *VlMYBA1* regulated a narrow set of genes involved in anthocyanin biosynthesis and identified novel genes associated with anthocyanin transport [280]. In another study, Zou et al. [285] focused on the development of transferable DNA markers for grapevine breeding and genetics. They developed a marker strategy targeting the Vitis collinear core genome and developed 2000 rhAmpSeq markers. They validated the marker panel in four biparental populations spanning the diversity of the *Vitis* genus, showing a transferability rate of 91.9% [285]. This marker development strategy has the potential to improve marker transferability in grapevine breeding.

Furthermore, the influence of transcription factors on grapevine biology and disease resistance has been investigated using grapevine transformed through the SE pathway. Guillaumie et al. [286] studied the role of the grapevine transcription factor *VvWRKY2* in cell wall structure and lignin biosynthesis. Transgenic tobacco plants overexpressing *VvWRKY2* exhibited alterations in lignin composition and expression of genes involved in lignin biosynthesis and cell wall formation [286].

In conclusion, research on transgenic grapevines involving Agrobacterium transformation, ZFN, TALEN, and CRISPR/Cas has explored various aspects of grapevine biology, including anthocyanin biosynthesis, sugar accumulation, disease resistance, and genetic factors influencing cell wall structure and lignin biosynthesis. Optimization of the CRISPR/Cas9 system using grape promoters has been shown to increase editing efficiency. Other genetic engineering approaches, such as biolistic co-transformation and cisgenic engineering, have also been explored in grapevine. These techniques offer targeted genome editing capabilities and have been used to edit specific genes of interest in grapevine, providing valuable insights into the potential applications of genetic engineering and marker-assisted breeding in grapevine improvement. These solutions achieved for trait improvement using genetic engineering techniques in grapevine would be challenging to achieve through traditional breeding methods.

## 6. Genetic Stability of Plants Regenerated from Somatic Embryos

SE is a method that offers the possibility of clonal plant regeneration. However, in some cases, plantlets regenerated via in vitro culture might develop altered characteristics and reveal a wide array of genetic variants. This variation can arise due to two phenomena: somaclonal variation and separation of chimeric layers.

### 6.1. Somaclonal Variation: Nemesis of Clonal Propagation, Ally in Crop Improvement

The path associated with the reprogramming of the cells of explants which leads to the formation of a callus, an apparently disorganized mass of cells, is accompanied by the appearance of somaclonal variations and other abnormalities, which, in turn, lead to plants with characteristics diverging from their mother plant. Some of the changes caused by somaclonal variation can be stable and therefore can be maintained, constituting a new source of genetic variability useful for breeding programs [135]. The most dominant hypothesis is that genetic instability is caused by stress to which the explant cells and the new cells generated in vitro are subjected. SE is usually achieved in vitro by exposing explants to PGRs and other treatments, which typically induces the formation of a callus—an apparently disorganized mass of cells that is considered the main source of somaclonal variation. These passages leading to callus formation often expose plant cells to stress and ultimately lead to alterations in the genome and epigenome [4]. It is thought that stress factors, applied only briefly, may lead to an increase in plant regeneration ability, as they lead to hormone redistribution. In particular, through creating the organizer cell niche by the initial accumulation of high auxin contents, the induction of histone hyperacetylation, and new auxin biosynthesis, induction has been observed. Reactive oxygen species (ROS) also play a positive role in stem cell induction and plant regeneration in vitro. In contrast, inhibition of ROS production or the use of ROS scavengers prevents the cell cycle and shoot regeneration [287].

Among the effects caused by in vitro culture stress are the occurrence of anomalous cytological events during callus formation and the prevalence of polyploidization and chromosome reduction events [36,288]. The presence of 2,4-D, one of the most used growth regulators in plant tissue culture, is considered the main agent responsible for this and other chromosomal abnormalities. Other factors, such as temperature variation or physical and chemical stresses, may contribute to chromosomal instability [289]. It has been observed that in explants of different plant species cultured in vitro, the initial events leading to SE appear to have undergone reprogramming of somatic cells to a gamete-like state, including chromosome segregation and the emergence of a haploid gamete-like cell appearance [288,290,291].

Although the production of haploids was initially considered a negative effect induced by the physiological and morphological disorders to which cells grown in vitro are subjected, today it constitutes an opportunity for innovative breeding strategies aimed at promoting and improving sustainable agriculture. The practical values of haploids in plant breeding have been illustrated by several authors, and therefore the in vitro switch from mitotic cell division to meiosis has aroused growing interest. Murray et al. [292] and De La Fuente et al. [293] introduced the concept of a cell-based in vitro breeding system (termed In Vitro Nurseries; IVNs). In IVNs, breeding cycle time could be substantially reduced by enabling rapid cell-level breeding cycles, without the need for flowering. The explants collected from mother plants could be cultured and brought to induce haploid cells after recombination without gametophyte development (artificial gametes). These cells can then be fused artificially in vitro [294]. This opportunity is of great interest, especially for crops like grapevine that have a long juvenile phase. In addition, IVNs will significantly reduce field management costs and environmental risks related to biotic and abiotic stresses. However, to apply IVNs more widely requires overcoming several bottlenecks. Cook et al. [294] distinguish three distinct phases: (i) in vitro production of haploid gamete-like cells inducing meiosis from somatic vegetative tissues; (ii) identifying/isolating artificial gametes carrying favorable alleles; and (iii) producing cell lines from selected artificial gametes followed by the fusion of selected artificial gametes to generate diploid cells as a starting point for the next generation in IVNs. Therefore, it is useful to develop efficient protocols to induce meiosis in vitro and regenerate haploid cell lines. Several substances added to culture media reduce chromosome number in cells maintained in vitro. Among these, chloramphenicol antibiotic treatment was shown to reduce chromosomes to a haploid state in root cells of barley seedlings [295]. Caffeine treatments have been used to induce somatic meiosis-like reductions in *Vicia* root tips [296]. The exogenous application of trichostatin A has also been used to induce the formation of haploid somatic embryos from male gametes of different species [297,298,299].

Somaclonal variation in grapevine can be phenotypically evaluated by observing morphological and physiological traits in ex vitro- and in vivo-grown plants, using the international standard descriptors (ampelographic and ampelometric) provided by the International Organization of Vine and Wine (OIV) [35]. However, the evaluation of the uniformity of morphological traits in field-grown plants is expensive, and it is necessary to wait several years to overcome the juvenile phase to evaluate fruit and wine characteristics. Moreover, some changes obtained after in vitro culture cannot be observed in planta, because differences that influence the biological activity may not affect the phenotype.

To complement the morphological characterization of regenerants in the field using the descriptors from the OIV, cytogenetic, biochemical, as well as DNA- and RNA-based technologies are sensitive tools, which can quickly provide information on genetic stability [36]. Cytogenetic studies on grapevine are often difficult, mainly due to the large number of small chromosomes and the difficulty of obtaining good chromosome preparations [271]. One of the most efficient techniques to detect different ploidy levels is based on flow cytometry. Different ploidy levels in grapevine regenerants via SE were detected by flow cytometry. Autotetraploid plants showed marked anatomical and morphological changes in shoots and mature leaves. Alterations have also been observed in stomata and chloroplast numbers, which were higher in tetraploids than in diploid mother plants. On the contrary, the stomatal index was markedly decreased in leaves of tetraploid regenerants [36]. Capriotti et al. [260] tested over 2300 ‘Ancellotta’ and ‘Lambrusco Salamino’ plants regenerated through SE for somaclonal variants for downy mildew (*Plasmopara viticola*) resistance after inoculation with a spore suspension. They identified 54 plants of ‘Lambrusco Salamino’ and 22 plants of ‘Ancellotta’ showing low levels of infection.

Different molecular markers have been used for the assessment of genetic fidelity of regenerants, most of which are based on PCR technology. The DNA markers most used for the verification of grapevine genetic fidelity are random amplified polymorphic DNA (RAPD), sequence-characterized amplified regions (SCARs), simple sequence repeats (SSRs), inter-simple sequence repeats (ISSRs), amplified fragment length polymorphisms (AFLPs), single-nucleotide polymorphisms (SNPs), expressed sequence tags (ESTs), and random amplified microsatellite polymorphisms (RAMPs) [47,221,300,301,302,303,304]. For an exhaustive and updated review of the molecular markers used for the assessment of genetic fidelity of in vitro regenerated plants, see Biswas and Kumar [305], and for grapevine, see Butiuc-Keul and Coste [271].

### 6.2. Chimerism in Grapevine and Segregation of Genotypes Through Somatic Embryogenesis

Chimerism refers to the presence of genetically distinct cell layers within a single plant [56,218,219,306]. This phenomenon has been observed in grapevines and has implications for cultivar identity, ancestry, and genetic improvement [218]. Chimerism can arise through somatic mutations that occur in one of the three meristematic cell layers in the apical meristem, which then differentiate into various plant tissues [307]. The existence of chimerism in grapevines has been demonstrated through DNA profiling using microsatellite loci. In some cases, more than two alleles have been observed at a locus, indicating the presence of chimerism [218].

Chimerism can manifest in various forms. Somatic chimerism occurs when different genetic lineages exist within different tissues of a grapevine plant. For example, a grapevine may have shoots or leaves with genetic characteristics true to the clone while its berries or flowers display characteristics different from the clone. This variation can arise due to genetic mutations or somatic hybridization events. Periclinal chimerism refers to the presence of different genetic lineages in distinct layers of tissue within a plant. It occurs when genetic mutations affect specific cell layers during plant development. This type of chimerism can result in variegated patterns of color or leaf morphology, where different tissue layers display different traits. They can be multiplied through grafting [218].

Chimerism in grapevines can have significant effects on phenotype and genetic diversity. Somatic mutations that give rise to chimeras can lead to morphological and agronomical differences, which can stabilize in grapevine plants and contribute to the genetic diversity of grapevine accessions [218,308]. This suggests that chimerism can modify phenotype and influence grapevine improvement through genetic transformation and conventional breeding strategies [218]. Chimerism significantly impacts grapevine clone stability by introducing genetic variability within a single plant. Periclinal chimeras, which consist of distinct cell layers (L1 and L2), can exhibit different phenotypes and genetic profiles, influencing both agronomic traits and cultivar identity. For instance, the study of the ‘Merlot’ cultivar revealed specific periclinal chimeras that could be propagated, suggesting potential for improved clonal selection [309]. However, SE, which typically regenerates plants from the L1 layer, may limit the expression of phenotypic diversity, as seen in the ‘Nebbiolo’ [56] and ‘Chardonnay 96’ [306] clones, where regenerated plants showed no significant phenotypic alterations compared to their parent plants. This indicates that while chimerism can enhance genetic diversity, the method of propagation can influence the stability and expression of these variations, highlighting a complex interplay between genetic chimerism and clonal propagation strategies [310].

Understanding chimerism is important for genetic diversity assessment, population structure analysis, and the development of new grapevine cultivars. Molecular markers, such as microsatellite loci and SNPs, can be used to investigate patterns of genetic diversity in grapevine germplasm collections [311]. These markers can provide reliable tools for characterizing the genetic diversity and population structure of grapevine accessions, including domesticated grapevine (*V. vinifera* ssp. *sativa*), wild relatives (*V. vinifera* ssp. *sylvestris*), interspecific hybrid cultivars, and rootstocks [311].

Chimerism in grapevines can have both positive and negative implications. As positive implications, chimerism can introduce unique traits or characteristics, and chimeras displaying desirable traits can be identified and propagated through clonal selection, allowing new grape clones with specific traits or improved quality to be produced. As negative implications, chimerism can lead to unpredictability in grapevine propagation, as the desired traits might not be stably inherited in subsequent generations. This can complicate breeding programs and commercial cultivation, and chimeric grapevines may produce fruits with varying characteristics, making it challenging to achieve uniformity in terms of flavor, color, or size.

## 7. Conclusions and Future Prospects

SE is a transformative biotechnological approach for grapevine improvement, germplasm conservation, and disease management. The ability of SE to regenerate whole plants from somatic cells offers substantial benefits, particularly in generating genetically uniform plants, preserving valuable grapevine germplasm, and enabling genetic transformation. Despite these benefits, SE’s practical application is constrained by challenges such as genotype recalcitrance, somaclonal variation, and difficulties in synchronizing embryo production. Addressing these challenges will be essential to fully realize SE’s potential in both research and commercial contexts. Nevertheless, SE remains an indispensable tool for grapevine research, providing solutions where traditional breeding methods fall short due to the grapevine’s long reproductive cycles and complex genetic makeup.

Future research efforts must focus on optimizing SE protocols to improve reproducibility across a broader range of grapevine genotypes. This will involve refining the selection of explants, adjusting culture medium compositions, and identifying optimal concentrations of PGRs and culture conditions. An important role will be played by the PGR-mediated epigenetic and chromatin modifications during SE induction and development. Recent investigations [4,17,312,313] on differentially expressed genes associated with chromatin modification have suggested that these modifications play unique roles in cell dedifferentiation and embryo development in several species, driving the transition of somatic cells into totipotent cell lines and subsequently in whole embryos through SE progression [314].

A deeper understanding of the mechanisms underlying somaclonal variation is critical to mitigating the genetic instability frequently observed in in vitro cultures. Furthermore, improving the synchronization of somatic embryo production is crucial for scaling up SE systems for commercial use. Further advances in the understanding of signaling pathways and molecular mechanisms that regulate embryogenesis could lead to significant improvements in this area.

The integration of SE with modern biotechnologies, such as genome editing tools (e.g., CRISPR/Cas9), RNA sequencing, and metabolomics, presents an exciting opportunity to enhance grapevine breeding programs. SE provides a foundational platform for precise genetic modifications, enabling the creation of grapevine cultivars with improved traits such as disease resistance, stress tolerance, and enhanced yield and quality.

Cryopreservation techniques also require further refinement to ensure high regeneration rates of conserved somatic embryos, particularly for the preservation of endangered species and elite cultivars. Additionally, the development of automated SE processes is essential for reducing costs and making SE more commercially viable. Automated systems for embryo culture, synchronization, and plantlet acclimatization will be instrumental in scaling up SE for large-scale applications.

While SE faces several challenges, ongoing research and refinement of SE techniques, coupled with the integration of emerging technologies, hold great promise for expanding its applications in grapevine research and production. By overcoming these limitations, SE has the potential to significantly enhance the resilience, sustainability, and adaptability of grapevine cultivation in response to the demands of modern horticulture and changing climate.

## Figures and Tables

**Figure 2 plants-13-03131-f002:**
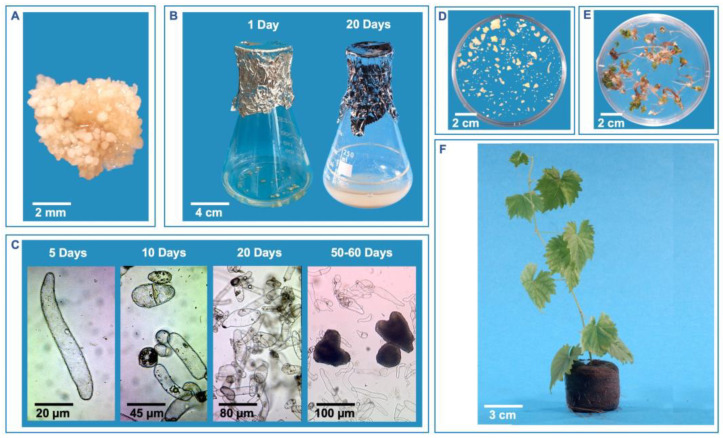
Development of somatic embryos and plantlets from cell suspension cultures of grapevine. (**A**) Proembryogenic masses (200–400 mg) are used for culture initiation. (**B**) Liquid cultures are maintained in 250 mL Erlenmeyer flasks containing 50 mL of liquid culture medium. (**C**) Images of cells growing in liquid culture and somatic embryos (globule and heart-shaped stages) differentiated after 40 days of initiation of culture. (**D**) Somatic embryos, collected by filtration after 2–3 months from the start of culture using a nylon mesh filter (2 mm), are incubated on growth regulator-free liquid medium. (**E**) Germination of the embryos occurs after approximately 30 days of culture on growth regulator-free solid medium. (**F**) The plantlets are acclimated in Jiffy pots and reach about 15 cm in height in about 40–60 days.

**Figure 3 plants-13-03131-f003:**
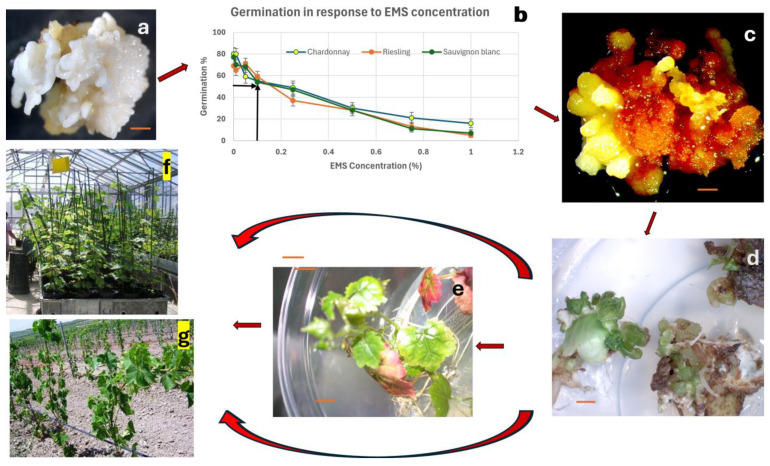
A scheme for using embryogenic cultures for mutation induction and screening. (**a**) Embryogenic culture establishment. (**b**) Optimizing mutagen dose through growth reduction studies [58]. (**c**) Development of somatic embryos after treatment with optimized mutagen dose. (**d**) Initial germination. (**e**) Screening the germinated embryos for the trait of interest in vitro. (**f**) and (**g**) Testing mutagenized populations in greenhouse and field conditions, respectively, for traits difficult to screen in vitro, such as bunch architecture, vine growth, fruit quality in table grapes, etc. Bars: (**a**,**c**,**d**) = 1 mm; (**e**) = 5 mm.

**Figure 4 plants-13-03131-f004:**
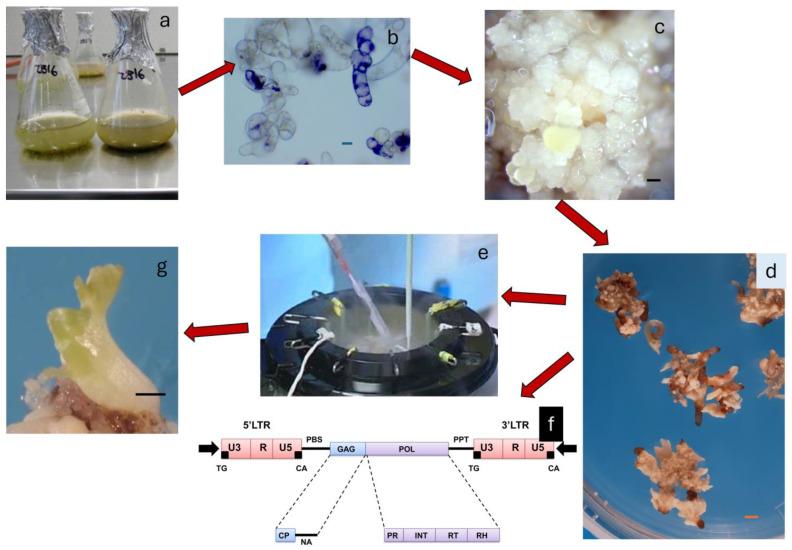
A scheme for transposon activation, tagging mutants and their recovery using embryogenic cultures of grapevine. (**a**) Cell culture established and subjected to stress for transposon activation. (**b**) Cells subjected to stress. (**c**) Somatic embryos (SEs) generated from stressed embryogenic cells. (**d**) Secondary SEs induced from primary SEs, with the clusters serially numbered. (**e**) Part of the labelled secondary SEs cryopreserved. (**f**) Another part of SEs subjected to RNAseq for transposon tagging. (**g**) Identified mutants of interest recovered from cryopreservation and regenerated. Bars: (**b**) = 10 µm; (**c**,**d**,**g**) = 1 mm.

## Data Availability

Not applicable.

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
