# Peer review of "Development and Applications of Somatic Embryogenesis in Grapevine (Vitis spp.)"

_plants, 2024, doi:10.3390/plants13223131_

Round 1
Reviewer 1 Report
Comments and Suggestions for Authors
This review described in details process of somatic embryogenesis in grape. Authors collected a large number of the publications and perform detailed analysis of each.
The text is well organized and written.
However, there are some points is still missing.
Namely, somatic embryo can be considered as endogenous auxin network, include multiply, cell fate specific auxin source, polarity induction and polar auxin transport, accompanied by local epigenetic re-programming specific for each cell type and position.
This part is rather missing and require more descriptions, at least in the part about perspectives.
Epigenetic aspects related with chromatin remodeling is a key in SE, and can serve as good markers were completely missing.
Some minor details:
Lines 36-37: please, explain what do you mean as fully mature differentiated cells? How can you have quantified this? Euchromatin entropy?
Function = fate.
Regeneration of damaged tissue? Do you mean de novo regeneration or tissue repair? There are many confusions in the field, please, clarify.
Line 45: “independent of the whole organism”? This is not true. Cell require carbon, hormone and other important molecule. Single cells became independent in vitro if they get carbon, hormoines, other nutrients.
Line 70: exogenous or endogenous PGR? Plant regeneration regulated exclusively by endogenous hormones, more precisely multidirectional transport of endogenous auxin with help of the other hormones as well.
Line 76: what is differentiation? Maybe cell fate re-establishment? On line 36 you already used differentiation with another meaning.
Lines 95 – 102: nice description. Why not to use euchromatin entropy as a marker of competent cell in grape? As well as capability of induction of de novo auxin synthesis?
Young organ fully fit this since produce auxin and have low euchromatin entropy.
Line 115: I would suggest to tell cell fate establishment as pro-vascular cell, right?
Line 140_ please, made a clear conclusion from 29 citation.
Line 145: “differentiated somatic embryos” - what do you mean as differentiated? Condense chromatin? High euchromatin entropy? Maybe “well established SE”?
Lines 141- 165: there are a lot of technical details. I would suggest to make this part shorter, describe only some basic point.
Line 193: do you mean “The embryogenic potential” in vitro? In planta all cultivars have almost the same embryogenic potential.
Line 214 – Expression does not necessary means involvement. It can be simple accompanied.
Lines 193 – 228: the basic (main) genes did not present here. The basic one are auxin biosynthesis one and auxin transport. The gene expression is only local, in few key cell, and can not be studied by “molecular methods”. I know, there are not so much information, but will be nice at least mention these genes. And chromatin modification genes, what, in turn, dependent form local auxin production/level.
Line 241: “seed-derived somatic embryos are not useful for clonal propagation” Why??
Figure 2: mµ?? Maybe µm??
Lines 244- 245: the key in somatic embryo induction is capability of the cell to produce endogenous auxin through certain pathway, and link with it euchromatin organocation. Did authors mean these parameters as physiological status? Please, clarify.
Lines 276- 289: it will be great to provide effect of exogenous hormones on endogenous one and as epigenetic regulator. Somatic embryogenesis regulated exclusively by inner hormones and it distribution/local response.
Lines 351- 374: it will be nice to make this part more structural and mention that long-term storage requires chromatin remodeling, in planta induce by ABA and can be induced in SE by certain kind of stress.
Line 671: “cisgenic”?
Lines 725 – 734: quite confusing part. The main source of s0maclonal variation is stage of unorganized callus with lacking epigenetic balance and ROS balance. This, in turn, induce genetic and epigenetic changes and involved in epigenetic memory maintenance.
Some details are there:
https://doi.org/10.3390/plants13020327
Comments on the Quality of English LanguageModerate polishing are required
Author Response
Reviewer 1
Comment 1
This review described in details process of somatic embryogenesis in grape. Authors collected a large number of the publications and perform detailed analysis of each.
Our comment / answer
Thank you for positively assessing our review.
Comment 2
The text is well organized and written.
Our comment / answer
Thanks again. Yes, all authors made an effort to improve the text before submission.
Comment 3
However, there are some points is still missing. Namely, somatic embryo can be considered as endogenous auxin network, include multiply, cell fate specific auxin source, polarity induction and polar auxin transport, accompanied by local epigenetic re-programming specific for each cell type and position. This part is rather missing and require more descriptions, at least in the part about perspectives.
Our comment / answer
We thank the reviewer for suggestions. We modified the MS as suggested by the Reviewer 1 by adding the following sentence in the "Conclusions and Future Prospects":
An important role will be played by the PGR-mediated epigenetic and chromatin modifications during somatic embryo (SE) induction and development. Recent investigations (Dal Santo et al. 2022; Salaün et al. 2021; Sivanesan et al. 2022; Wang et al. 2020) on differentially expressed genes associated with chromatin modification have suggested that these modifications play unique roles in cell dedifferentiation and embryo development in several species driving the transition of somatic cells into totipotent cell lines and subsequently in whole embryos through SE progression (Ramakrishnan et al. 2023).
Additionally, we mention about the effect of epigenetic changes and abscisic acid on the synchronisation and germination of somatic embryos in Section 3 (Lines 211-214).
Comment 4
Epigenetic aspects related with chromatin remodeling is a key in SE, and can serve as good markers were completely missing.
Our comment / answer –
We thank the reviewer for suggestions. We modified the MS as suggested by the Reviewer 1 by adding the following sentence in the "Introduction”:
Despite the fact that all the diploid cells in a single individual have the same genomic DNA, different cell types have distinct cell characteristics, and only some cells are to-tipotent to become an embryo. This discrepancy suggests that this difference in the ability to generate somatic embryos is not driven by simple DNA sequence, but probably by the different epigenetic changes of different loci of totipotent cells. In particular, totipotent cells differ from their surrounding somatic cells mainly in five respects: large nucleus, large nucleolus, fragmented vacuoles, symplasmic isolation and low heterochromatin content (Godel-Jedrychowska et al. 2020; Verdeil et al. 2007; Wang et al. 2022). Recent studies have shown that epigenetic aspects related with chromatin remodeling play a key role in SE, and can serve as good markers (Peng et al. 2023).
Lines 61-69.
Comment 5
Some minor details:
Lines 36-37: please, explain what do you mean as fully mature differentiated cells? How can you have quantified this? Euchromatin entropy?
Function = fate.
Regeneration of damaged tissue? Do you mean de novo regeneration or tissue repair? There are many confusions in the field, please, clarify.
Our comment / answer
We thank the reviewer for suggestions. We modified the MS as suggested by the Reviewer. In the new MS, we have reformulated the sentence being more specific on points raised by the Reviewer:
The early observations on the capacity that plants have to react to tissue injury by leading fully differentiated somatic cells to change their fate, thus favouring the formation of unorganized cell mass, called callus, which play a prominent role in damaged tissue, but also capable of regenerating new organs, led to the pioneering studies of plant tissue and cell culture in vitro.
Lines 36-40.
Comment 6
Line 45: “independent of the whole organism”? This is not true. Cell require carbon, hormone and other important molecule. Single cells became independent in vitro if they get carbon, hormoines, other nutrients.
Our comment / answer
We thank the reviewer for the observation which is correct and we agree. In fact, it is true that the single cell is not independent from the whole organism. However, we reported the hypothesis that Haberlandt formulated in 1925 (Zelle und Elementaroigan, - Biol, Zentralbl, 45: 257-272). We have reformulated the sentence by reporting Haberlandt's original sentence in German to highlight that this is his hypothesis. We hope that the new version is clearer:
Haberlandt hypothesized that a single cell is a living unit, an individual in itself that is to some extent independent of the whole organism: ‘Als Elementar Organismus ..... ist die Zelle eine Lebenseinheit, ein Individuum fiir sich, das ein vom Gesamtorganismus his zu einem gewissen Grade unabhangiges Eigenleben fuhrt’ (Haberlandt 1925).
Lines 46-48.
Comment 7
Line 70: exogenous or endogenous PGR? Plant regeneration regulated exclusively by endogenous hormones, more precisely multidirectional transport of endogenous auxin with help of the other hormones as well.
Our comment / answer
We thank the reviewer for the observation. We apologize for not being clear, the exogenous stimuli do not refer to the PGR but to the stresses (physical and chemical) to which the explants are subjected to stimulate the regeneration of somatic embryos. We have modified the sentence. We hope that the new version is clearer:
During this process of dedifferentiation and differentiation of plant cells, the explant responds not only to endogenous but also to exogenous stimuli (including different types of stress: physical injury, low or high temperature, heavy metals, osmotic and drought stress), which trigger the induction of a signaling response and, consequently, can often profoundly modify the cell fate.
Lines 84-86.
Comment 8
Line 76: what is differentiation? Maybe cell fate re-establishment? On line 36 you already used differentiation with another meaning.
Our comment / answer
We thank the reviewer for the observation. We agree that the sentence can be confusing. We have replaced "differentiation" with "process". We hope that the new version is clearer:
....... and subsequently will be translated into proteins involved in the process that ultimately will lead to the regeneration of a new somatic embryo.
Lines 94-96.
Comment 9
Lines 95 – 102: nice description. Why not to use euchromatin entropy as a marker of competent cell in grape? As well as capability of induction of de novo auxin synthesis?
Young organ fully fit this since produce auxin and have low euchromatin entropy.
Our comment / answer
Thank you for the suggestion and we agree with your hypothesis. We have added this sentence to suggest this.
The fact that floral buds of plants have high endogenous levels of auxins [29] and highly dynamic DNA methylation states [30] during their development suggest that correct timing of explant harvest is critical for SE induction in grapevine, considering most successful protocols are based on the use of flower buds (Table 1).
Comment 10
Line 115: I would suggest to tell cell fate establishment as pro-vascular cell, right?
Our comment / answer -
We fail to understand this comment, sorry. If you mean that the somatic embryos have no vascular connection to maternal tissue, it is understood when we write “Globular embryos usually appear on the surface of the embryogenic calli” in the next line (now Line 150). But to make it clearer, we have added “…and have no vascular connections” – Line 151.
Comment 11
Line 140_ please, made a clear conclusion from 29 citation.
Our comment / answer
We have reformulated the sentence. We hope that the new version of the MS is clearer:
The authors hypothesize that the presence of a persistent suspensor, typical of somatic embryos regenerated in liquid media, modulates development, ultimately resulting in rapid germination and a high plant-regeneration rate [123].
Now lines 172-176.
Comment 12
Line 145: “differentiated somatic embryos” - what do you mean as differentiated? Condense chromatin? High euchromatin entropy? Maybe “well established SE”?
Our comment / answer – Here we describe the morphological development stage of somatic embryos. As we reduced this section in agreement with your comment No 13, this part was totally removed.
Comment 13
Lines 141- 165: there are a lot of technical details. I would suggest to make this part shorter, describe only some basic point.
Our comment / answer
We agree that there is too much detail in this section. We have now reduced it to few sentences.
Jayasankar et al. [124] observed that the embryogenic suspension cultures of V. vinifera ‘Thompson Seedless’ do not progress beyond heart shape when the auxin 2,4-dichlorophenoxyacetic acid (2,4-D) was removed from the medium while those of ‘Chardonnay’ developed fully. To synchronise the development of somatic embryos, they inoculated < 960 µm fraction into media without auxin. Sieving of proembryogenic masses (PEMs) and subculture resulted in the synchronization of embryo development and reduced browning and abnormalities such as fasciation or fusion during differentiation. As a result, they achieved 60% conversion of SEs into plants.
Comment 14
Line 193: do you mean “The embryogenic potential” in vitro? In planta all cultivars have almost the same embryogenic potential.
Our comment / answer
It is true that in many species all cultivars have almost the same embryogenic potential. However, in grapevine the genotype is the major determining factor, with embryogenic response varying from 0.1 to 5.1%, about 50-fold difference (see Carra et al. 2016 Scientia Horticulturae 204 (2016) 123–127). We have reformulated the sentence:
Although in many species all cultivars have almost the same embryogenic potential, in grapevine SE competence is strongly genotype dependent. The embryogenic potential of cultivars varies considerably, in a comparison of eight grapevine cultivars about 50-fold difference was observed (Carra et al. 2016), and although multiple methods have been published, for certain cultivars the technique still needs further improvement [33,133].
Comment 15
Line 214 – Expression does not necessary means involvement. It can be simple accompanied.
Our comment / answer
Please see answer to comment number 16 as these are related.
Comment 16
Lines 193 – 228: the basic (main) genes did not present here. The basic one are auxin biosynthesis one and auxin transport. The gene expression is only local, in few key cell, and can not be studied by “molecular methods”. I know, there are not so much information, but will be nice at least mention these genes. And chromatin modification genes, what, in turn, dependent form local auxin production/level.
Our comment / answer
Many thanks for getting us to include this part of the story. We have added a major section on auxin signalling and chromatin modification and involvement of various genes and TFs:
Auxin plays a pivotal role as auxin gradients influence cell fate and differentiation. In grapevine, the genes VvPIN and VvAUX1/LAX are critical for creating these auxin gradients. VvPIN genes encode auxin efflux carriers that facilitate the directional transport of auxin out of cells, while VvAUX1 and LAX genes encode auxin influx carriers that mediate the uptake of auxin into cells. The interplay between these transporters is essential for maintaining auxin homeostasis and creating localized auxin maxima that drive development and organ formation [135-137]. The model of the PIN efflux system-dependent auxin gradients in the control of growth and patterning in zygotic embryogenesis as reviewed by Elhiti and Stasolla [138] applies to SE as well. Exogenous application of auxin, a critical requirement for SE, triggers a general reprogramming of somatic cell transcriptomes and modulates the expression of many SE-associated transcription factors. High concentrations of auxin in culture results in a significant increase in the size of nucleus, suggesting a reorganization of the chromatin [139]. This stress leads to modification of chromatin state which in turn results in the activation of transcription factors such as WUS, LEC genes or BBM that are specific to embryogenic programs [2,140]. These results suggest that hormones, especially auxin, trigger a general reprogramming of gene expression through chromatin modifications and activation of specific transcription factors.
Comment 17
Line 241: “seed-derived somatic embryos are not useful for clonal propagation” Why??
Our comment / answer
Sorry for the mistake, we meant to say that somatic embryos obtained from in vitro seed culture are not useful for clonal propagation because of the possible genetic contamination of the callus obtained from the zygotic embryo that have a different genome than the mother plant. We have modified the sentence:
However, zygotic seed-derived somatic embryos are not useful for clonal propagation.
Comment 18
Figure 2: mµ?? Maybe µm??
Our comment / answer
We apologize for the mistake; we have modified Fig. 2 in the revised version of the MS. As it is confusing, we have removed the previous version (journal requires us to leave ‘track changes’ when revising.
Comment 19
Lines 244- 245: the key in somatic embryo induction is capability of the cell to produce endogenous auxin through certain pathway, and link with it euchromatin organocation. Did authors mean these parameters as physiological status? Please, clarify.
Our comment / answer
We agree that this statement can be confusing. By physiological status, we meant development stage which is not accurate. We have read the two publications and modified the statements to reflect the research results of the authors. Now it reads:
It is widely known that the developmental stage of the explant affects the effectiveness of SE induction, and the exogenous auxin application in right concentration and its type and duration of incubation also have considerable influence on the success of the protocol [79,148].
Comment 20
Lines 276- 289: it will be great to provide effect of exogenous hormones on endogenous one and as epigenetic regulator. Somatic embryogenesis regulated exclusively by inner hormones and it distribution/local response.
Our comment / answer
We appreciate this suggestion and have included a paragraph on this relationship.
Exogenous application of auxin results in modification in endogenous auxin signaling. For example in Coffea canephora, there is an increase in the content of endogenous IAA and in the expression of the genes that code for the enzyme tryptophan aminotransferase that are involved in the biosynthesis of IAA ([161]. While facilitating SE induction, exogenously applied synthetic PGR can also disrupt the endogenous auxin balance and its transport that can result in abnormalities in somatic embryos. This problem has been highlighted in a recent review by Garcia et al. [130], focusing on 2,4-D mediated abnormalities. Using V. vinifera cv. ‘Chardonnay’ cotyledonary embryos with distinct morphologies as a model system, Ya et al. [162] demonstrated that the cellular concentrations of IAA and ABA were significantly higher in normal cotyledonary embryos compared to vitrified, fused or elongated cotyledonary embryos. Comparative transcriptome analysis also revealed significant differences in gene expression of the hormone signaling pathways in normal and abnormal cotyledonary embryos of grapevine providing further evidence of the importance of regulation of endogenous auxin and abscisic acid for the production of healthy and viable somatic embryos through proper regulation of exogenously applied auxin.
Comment 21
Lines 351- 374: it will be nice to make this part more structural and mention that long-term storage requires chromatin remodeling, in planta induce by ABA and can be induced in SE by certain kind of stress.
Our comment / answer
We have mentioned chromatin remodeling in response to other comments by the same reviewer. To complete the storage part, we have added one sentence:
Although the involvement of ABA metabolism in the maturation of grapevine somatic embryos has been demonstrated as already described {Acanda, 2020 #3751}, research is lacking on the use of various stress factors stimulating endogenous ABA biosynthesis during the process, thus improving maturation allowing better storage of somatic embryos.
Comment 22
Line 671: “cisgenic”?
Our comment / answer
The term “cisgenic” is correct (Dhekney, S.A.; Li, Z.T.; Gray, D.J. Grapevines engineered to express cisgenic Vitis vinifera thaumatin-like protein exhibit fungal disease resistance. In Vitro Cellular & Developmental Biology-Plant 2011, 47, 458-466).
Comment 23
Lines 725 – 734: quite confusing part. The main source of somaclonal variation is stage of unorganized callus with lacking epigenetic balance and ROS balance. This, in turn, induce genetic and epigenetic changes and involved in epigenetic memory maintenance.
Some details are there:
https://doi.org/10.3390/plants13020327
Our comment / answer
We have reformulated the sentence. We hope that the new version of the MS is clearer:
The path associated with the reprogramming of the cells of the explants which leads to the formation of the callus, an apparently disorganized mass of cells, is accompanied by the appearance of somaclonal variations and other abnormalities, which, in turn, lead to plants with characteristics diverging from their mother plant. Some of the changes caused by somaclonal variation can be stable and therefore can be maintained constituting a new source of genetic variability useful for breeding programs [134]. The most dominant hypothesis is that genetic instability is caused by stress to which the explant cells and the new cells generated in vitro are subjected. SE is usually achieved in vitro by exposing plant explants to PGR and other treatments which induces the typical formation of a callus; an apparently disorganized mass of cells that is considered the main source of somaclonal variation. These passages leading to callus formation often expose plant cells to stress and ultimately lead to unwanted alterations in the genome and epigenome [2]. It is thought that stress factors, applied only briefly, may lead to an increase in plant regeneration ability as they lead to hormone redistribution. In particular, through creating the organizer cell niche by the initial accumulation of high auxin contents, the induction of histone hyperacetylation, and new auxin biosynthesis induction was observed. ROS also play a positive role in stem cell induction and plant regeneration in vitro. In contrast, inhibition of ROS production or the use of ROS scavengers prevents cell cycle and shoot regeneration (Pasternak and Steinmacher 2024).
Comment 24
Comments on the Quality of English Language
Moderate polishing are required
Our comment / answer
All the four authors and a colleague with English as forst language have gone through the manuscript again and made changes to improve the text.
Reviewer 2 Report
Comments and Suggestions for Authors
Development and applications of somatic embryogenesis in 2
grapevine (Vitis spp.)
Angela Carra, Akila Wijerathna-Yapa, Ranjith Pathirana and Francesco Carimi
Some specific comments and the line number in which they are found are listed below.
The English of the manuscript requires minor revision. Please see below for a couple of comments on the abstract suggested by the reviewer.
Is there any reason the authors did not consider including a table with the information generated on somatic embryogenesis in Vitis spp? Given the nature of the review, it could be beneficial.
General observation. The manuscript provides valuable information. However, the way it is presented is difficult to assimilate and understand. It is a large manuscript, more than 12,000 words. Several of the paragraphs exceed 450 words and contain a large amount of technical information. For this reason, grammatical errors can be seen in the text.
Lines 46 & 47. This sentence needs to be changed. First, it was three groups, independently, between late 1957 and early 1959, who discovered the SE process, and it was Reinert’s group who used the term “adventive embryos” for the first time in their second publication. As shown in Figure 4, Steward observed “the beginning of organization in a nodule leading to root formation” and the “emergence of roots from tissue cultures” in Figures 5 & 6. He never observed embryogenic structures.
ü Waris H., A striking morphogenetic effect of amino acid in seed plant, Suom Kemistil, 30B(5-6):121, (1957).
ü Steward F. C. et al., Growth and organized development of cultured cells. II. Organization in cultures grown from freely suspended cells, Am. J. Bot., 45(10):705-708, (1958).
ü Reinert J., Untersuchungen uber die Morphogenese an Gewebellulturen Ber. Dtsch. Bot. Ges. 71, 15, (1958). [Studies on the morphogenesis of tissue cultures]
ü Reinert J., Uber die kontrolle der morphogenese und die induktion von adventivembryonen an gewebekulturen aus karotten, Planta, 53(3): 318-333, (1959). [On the control of morphogenesis and the induction of adventive embryos in carrot tissue cultures].
On the other hand, Haberlant's hypothesis was first demonstrated in a couple of publications in the journal Science, as part of Vilma Vasil's doctoral thesis under the direction of Hildebrand at the University of Wisconsin.
ü Vasil V, Hildebrand AC (1965) Differentiation of tobacco plants from single, isolated cells in micro cultures. Science 150 (3698):889-892.
ü Vasil V, Hildebrandt AC (1965) Growth and tissue formation from single, isolated tobacco cells in microculture. Science 147 (3664):1454-1455.
Line 55. The authors use the terms “phytohormone,” “hormonal,” and Plant Growth Regulator (line 2777) interchangeably throughout the manuscript. Without entering into the controversy of whether the term is correct (in particular, this reviewer thinks it is misused and the term plant growth regulator should be used), the authors should standardize the use of the terms throughout their manuscript.
ü Bennett T, Leyser O (2014) The auxin question: A philosophical overview. In: Zazimalová E, Petrášek J, Benková E (eds) Auxin and Its Role in Plant Development. Springer, Vienna, pp 3-19.
Lines 205 and 206. The nomenclature for naming genes is incorrect. It should be corrected throughout the manuscript, SERK, not SERK.
Lines 278, 271. The term benzylaminopurine is incorrect. Adenine is a purine, so it should not be repeated in the name. The correct name is benzyladenine (BA). This should be corrected.
Lines 328 – 421. This section is difficult to read. It provides much information in three lengthy paragraphs. This reviewer suggests that the authors revise these three large paragraphs to make the information presented more accessible.

Please, see the report.
Author Response
Reviewer 2
Comment 1
Is there any reason the authors did not consider including a table with the information generated on somatic embryogenesis in Vitis spp? Given the nature of the review, it could be beneficial.
Our comment / answer
We thank the reviewer for the suggestion. In the revised version of the MS we have added the new table as suggested by the Reviewer 2. It goes as Table 1. This has added many more references.
Comment 2
General observation. The manuscript provides valuable information. However, the way it is presented is difficult to assimilate and understand. It is a large manuscript, more than 12,000 words. Several of the paragraphs exceed 450 words and contain a large amount of technical information. For this reason, grammatical errors can be seen in the text.
Our comment / answer
We thank the Reviewer for the comments and suggestions, we have revised the MS and tried to make sentences shorter. However, when addressing the many comments, particularly Reviewer 1, the length of text increased. As it is a comprehensive review, we hope that the length is acceptable and the new version is easier to understand now.
Comment 3
Lines 46 & 47. This sentence needs to be changed. First, it was three groups, independently, between late 1957 and early 1959, who discovered the SE process, and it was Reinert’s group who used the term “adventive embryos” for the first time in their second publication. As shown in Figure 4, Steward observed “the beginning of organization in a nodule leading to root formation” and the “emergence of roots from tissue cultures” in Figures 5 & 6. He never observed embryogenic structures.
Waris H., A striking morphogenetic effect of amino acid in seed plant, Suom Kemistil, 30B(5-6):121, (1957).
Steward F. C. et al., Growth and organized development of cultured cells. II. Organization in cultures grown from freely suspended cells, Am. J. Bot., 45(10):705-708, (1958).
Reinert J., Untersuchungen uber die Morphogenese an Gewebellulturen Ber. Dtsch. Bot. Ges. 71, 15, (1958). [Studies on the morphogenesis of tissue cultures]
Reinert J., Uber die kontrolle der morphogenese und die induktion von adventivembryonen an gewebekulturen aus karotten, Planta, 53(3): 318-333, (1959).
[On the control of morphogenesis and the induction of adventive embryos in carrot tissue cultures].
On the other hand, Haberlant's hypothesis was first demonstrated in a couple of publications in the journal Science, as part of Vilma Vasil's doctoral thesis under the direction of Hildebrand at the University of Wisconsin.
Vasil V, Hildebrand AC (1965) Differentiation of tobacco plants from single, isolated cells in micro cultures. Science 150 (3698):889-892.
Vasil V, Hildebrandt AC (1965) Growth and tissue formation from single, isolated tobacco cells in microculture. Science 147 (3664):1454-1455.
Our comment / answer
We thank the Reviewer for the suggestion. In the revised version of the MS we have added the information provided by Reviewer 2:
Direct evidence supporting the hypothesis that it was possible to regenerate plant organs in vitro has been lacking until the end of the 1950s. Three independent groups, between 1957 and 1958, discovered the regeneration process in Oenanthe aquatica (Waris 1957) and Daucus carota (Reinert 1958, Steward et al. 1958). It was Reinert who used the term ‘adventive embryos’ for the first time in 1959 (Reinert 1959). Finally, Haberlandt's hypothesis of producing a whole plant from a single cell was demonstrated in the mid-1960s in two publications as part of Vilma Vasil's doctoral thesis under the direction of Hildebrand at the University of Wisconsin thus demonstrating the totipotency of plant cells (Vasil and Hildebrand 1965a; Vasil and Hildebrand 1965b).
Comment 4
Line 55. The authors use the terms “phytohormone,” “hormonal,” and Plant Growth Regulator (line 2777) interchangeably throughout the manuscript. Without entering into the controversy of whether the term is correct (in particular, this reviewer thinks it is misused and the term plant growth regulator should be used), the authors should standardize the use of the terms throughout their manuscript.
Bennett T, Leyser O (2014) The auxin question: A philosophical overview. In: Zazimalová E, Petrášek J, Benková E (eds) Auxin and Its Role in Plant Development. Springer, Vienna, pp 3-19.
Our comment / answer –
We thank the Reviewer for the suggestion, and we agree that plant growth regulator is the better word in some sentences. We have revised the MS as suggested. However, internal hormone signalling is widely used and in this context we would like to keep this combination of words as PGR signalling is never used.
Comment 5
Lines 205 and 206. The nomenclature for naming genes is incorrect. It should be corrected throughout the manuscript, SERK, not SERK.
Our comment / answer
We have revised the MS as suggested by the Reviewer. Where reference is made to the gene, we have italicised.
Comment 6
Lines 278, 271. The term benzylaminopurine is incorrect. Adenine is a purine, so it should not be repeated in the name. The correct name is benzyladenine (BA). This should be corrected.
Our comment / answer
We agree with this view. We have revised the MS as suggested by the Reviewer.
Comment 7
Lines 328 – 421. This section is difficult to read. It provides much information in three lengthy paragraphs. This reviewer suggests that the authors revise these three large paragraphs to make the information presented more accessible.
Our comment / answer
We have revised the paragraphs and reduced the text where possible. Another sentence had to be added in response to a comment by Reviewer 1. We hope the current version reads better.
The authors a grateful to the reviewer for suggestions to further improve the manuscript .
Round 2
Reviewer 1 Report
Comments and Suggestions for Authors
Thank you very much for clear answers.
The text is partially improve, but some basic points is missing.
Authors still focus on exogenous hormone, but not endogenous one. In classical SE model include activation of endogenous auxin biosynthesis follow by induction of polarity, cell fate etc. The main focus should be on these points.
Please, add these points.
There are some details, but there are more. Moreover, some senteces are too long and confusing.
Line 46: „fully mature differentiated somatic cells to change their function/fate“ can not change fate: only partially differentiated cells can perfrom chromatin remodelling, activate auxin synthesis and change fate.
Lines 83 -87: long sentence, with confusing conlusion. Endogenous hormones (auixn is only one which can build polarity) are responsible for SE.
Lines 91- 96: very long sentence, with confusing conclusions. The first step in SE is induction of endogenous auxin synthesis with TAA1/TAR2 pathway.
Exogenous hormone play role as primary effect on endogenous auxin synthesis and further distribution. Yes, there are a lot of genes expressed thereafter, but this is not a primary.
Lines 121 – 124: the influoresence have multiply auxin biosynthesis pathways and flexible epigenetic modification. This is a key. Other characteristics are secondary only.
Figure 3, 4 – no scale bar.
Author Response
Reviewer 1
Comment 1
Thank you very much for clear answers
Our comment / answer
Thank you for positively assessing our revision.
Comment 2
The text is partially improve, but some basic points is missing.
Authors still focus on exogenous hormone, but not endogenous one. In classical SE model include activation of endogenous auxin biosynthesis follow by induction of polarity, cell fate etc. The main focus should be on these points.
Please, add these points.
Our comment / answer
We thank the reviewer for suggestions. We modified the MS as suggested by the Reviewer 1 by adding the following sentences in Introduction
“During this process of dedifferentiation and differentiation of plant cells, the explant responds not only to endogenous but also to exogenous stimuli (including different types of stress), which modify the endogenous hormonal balance. The evidence supports the notion of a major role of auxins in the establishment of polarity and embryo initiation and development [28-32].. For the understanding of this important plant regeneration model, the interactions between the different PGR, mainly auxins, cytokinins (CKs), ethylene and abscisic acid (ABA), during the induction of SE are of fundamental importance [33]. In particular, it has been observed that the induction of the auxin biosynthesis genes TAA1/TAR2, an increase of cellular auxin concentration, and its polar transport are required for cell reprogramming and embryo regeneration [29,30]. “
as well as in the newly revised Section 4.2.
“During the early stages of embryogenesis, endogenous auxin levels rise significantly, a phenomenon that is associated with the activation of stress signaling pathways and alterations in chromatin structure. Several studies have demonstrated that exogenously applied auxins elevate the levels of endogenous IAA in explants undergoing SE. While the induction of embryo identity in somatic explants does not depend on endogenous auxin biosynthesis, maintaining embryo identity requires an increase in endogenous auxin levels. This elevation, along with proper auxin transport, is crucial for promoting the differentiation of embryonic cells into histo-differentiated somatic embryos [176]. It has also been reported that 2,4-D promotes the production of IAA-binding proteins, enhancing the sensitivity of cells to IAA and rendering them competent for embryogenesis [177]. Thus, the accumulation of endogenous auxin is crucial for altering cell fate and laying the foundation for embryogenic processes [178].”
To make the Review more orderly, we have moved “Stages of SE” from Section 2 to Section 3, retaining only Explants in Section 2.
We have also changed the title of Section 4.2 from “Other Factors Controlling SE” to “External Factors Controlling SE”. We want to highlight the different external factors researchers can control to optimize SE process. We hope that these changes and additional information on endogenous auxin, chromatin remodeling and epigenetic modifications make the distinction clearer.
This change also resulted in moving some duplicated information on explants from Section 4.2 to Section 2. We have also further revised the manuscript in many other sections to improve clarity and the language.
Comment 3
There are some details, but there are more. Moreover, some senteces are too long and confusing.
Line 46: fully mature differentiated somatic cells to change their function/fate“ can not change fate: only partially differentiated cells can perfrom chromatin remodelling, activate auxin synthesis and change fate.
Our comment / answer
We thank the reviewer for suggestions. We modified the MS as suggested by the Reviewer 1 by adding the sentence in the "Introduction”:
“The early observations on the capacity that plants have to react to tissue injury by leading fully partially differentiated somatic cells to change their fate, thus ……….”
Comment 4
Lines 83 -87: long sentence, with confusing conlusion. Endogenous hormones (auixn is only one which can build polarity) are responsible for SE.
Our comment / answer
We thank the reviewer for suggestions. We modified the MS as suggested by the Reviewer:
During this process of dedifferentiation and differentiation of plant cells, the explant responds not only to endogenous but also to exogenous stimuli (including different types of stress), which modify the endogenous hormonal balance. The evidence supports the notion of a major role of auxins in the establishment of polarity and embryo initiation and development (Souter and Lindsey 2000).
Comment 5
Lines 91- 96: very long sentence, with confusing conclusions. The first step in SE is induction of endogenous auxin synthesis with TAA1/TAR2 pathway.
Exogenous hormone play role as primary effect on endogenous auxin synthesis and further distribution. Yes, there are a lot of genes expressed thereafter, but this is not a primary.
Our comment / answer
We thank the reviewer for the observation. As suggested, we have removed the confusing sentence and replaced it with the following, emphasizing the role of TAA1/TAR2:
“In particular, it has been observed that the induction of the auxin biosynthesis genes TAA1/TAR2, an increase of cellular auxin concentration, and its polar transport are required for cell reprogramming and embryo regeneration (Rodríguez-Sanz et al. 2015; Pérez-Pérez et al. 2019).”
Comment 6
Lines 121 – 124: the influoresence have multiply auxin biosynthesis pathways and flexible epigenetic modification. This is a key. Other characteristics are secondary only.
Our comment / answer
We thank the reviewer for the observation. We have added three key references on auxin biosynthesis and epigenetic modification during flower development and added a few sentences to describe this and connected it with grapevine SE using floral explants. We hope that the new version is clearer and more accurate:
“In grapevine, the best results are usually obtained with explants of floral origin such as whole flowers, anthers, filaments, stigmas/styles, ovaries and pistils (Figure 1 A-B, Table 1). The presence of multiple pathways of auxin biosynthesis within the inflorescence makes this organ an elegant model for studying SE. High-precision dynamic spatiotemporal auxin gradients within the inflorescence meristem are coordinated with growth [29,30]. This ensures that cells are exposed to a high level of auxin over time to activate organogenesis. As floral meristems are initiated in the axils, the timing and duration of exposure of cells to high auxin is governed temporally within the tissue [30]. This makes the floral organs very sensitive to the presence of exogenous PGR. The other key factor is epigenetic modifications. During the transition from the vegetative to the reproductive phase of development, the patterns of DNA and histone methylation change significantly [31].”
Comment 7
Figure 3, 4 – no scale bar.
Our comment / answer
Thank you for noting the lapse. We have now added scale bars in Figures 3 and 4. Additionally, we have given error bars for Fig. 3b as well.
Reviewer 2 Report
Comments and Suggestions for Authors
None
Author Response
Dear Reviewer 2, Thank you for accepting our First Revision. After this revision we believe the Review has further improved.
Round 3
Reviewer 1 Report
Comments and Suggestions for Authors
Thank you! Everything is fine now! My best regards!